# PLS-BASED APPROACH FOR FAIR REPRESENTATION LEARNING

## ABSTRACT

We revisit the problem of fair representation learning by proposing Fair Partial Least Squares (PLS) components. PLS is widely used in statistics to efficiently reduce the dimension of the data by providing representation tailored for the prediction. We propose a novel method to incorporate fairness constraints in the construction of PLS components. This new algorithm provides a feasible way to construct such features both in the linear and the non linear case using kernel embeddings. The efficiency of our method is evaluated on different datasets, and we prove its superiority with respect to standard fair PCA method.

## 1 INTRODUCTION

Over the past few years, the increasing use of automated decision-making systems has been widely installed in businesses of private companies of all types, as well as government applications. Since many of these decisions are made in sensitive domains, including healthcare (Morik, 2010), finance (Trippi & Turban, 1992), criminal justice (Angwin et al., 2016), or hiring (Dastin, 2018), society has experienced a significant impact on people's lives. This fact has made the intersection between Artificial Intelligence (AI), Ethics and Law a crucial area of current research. Despite the success demonstrated by Machine Learning (ML) in these decision-making processes, there is a growing concern regarding the potential discriminatory biases in the decision rules.

One promising approach to mitigate unfair prediction outcomes is fair representation learning proposed by Zemel et al. (2013) (see Section 2 for related work), which seek to learn meaningful representations that maintain the content necessary for a particular task while removing indicators of protected group membership. Once the fair representation is learned, any prediction model constructed on the top of the fair representation (i.e. using the representation as an input vector) are expected to be fair. Several works related to ours, such as Kleindessner et al. (2023); Olfat & Aswani (2019); Lee et al. (2022a), tackle the objective using principal component analysis (PCA). However, the new representation tends to be less useful for predicting the target when it is strongly correlated with some directions in the data that have low variance.

We propose an alternative formulation based on the dimensionality reduction statistical technique Partial Least Squares (PLS) (Bair et al., 2006). The paper introduces fairness for PLS as doing PLS while minimizing the dependence of the projections with the demographic attribute. The main objective is to learn a representation that can trade-off some measure of fairness (e.g. statistical parity, equal opportunity) with utility (e.g. covariance with respect to the target, accuracy) and can be kernelized. Specifically, the goal is to create a representation that: (i) has lower dimension; (ii) preserves information about the input space; (iii) is useful for predicting the target; (iv) is approximately independent of the sensitive variable. Our formulation has the same complexity as standard Partial Least Squares, or Kernel Partial Least Squares, and have applications on different domains and with different data structures as tabular, image or text embeddings. To sum up, we address the following questions: (1) How can fairness be defined in the context of PLS? And (2) How to integrate feasible fairness constraints into PLS algorithms?

**Outline of Contributions.** We make both theoretical and practical contributions in the field of fair representation learning and fair machine learning by proposing a dimensionality reduction framework for fair representation. More precisely, our contributions can be outlined as follows.

- Fair Partial Least Squares: in section 3 we first review the Partial Least Squares method and then we propose the fair formulation as a regularization in the iterative process of obtaining the weights.

- Kernel Fair Partial Least Squares: in section 4.1 we present how the method of Fair PLS can be extended to the non linear case by using kernel embedding and the Hilbert Schmidt independence criterion (HSIC) as the fairness term.

- Application to different fields and different data: we present diverse experiments in both tasks (classification and regression) for tabular data and we discuss how such framework could improve fairness for Natural Language Processing algorithms. Some details and experiments are deferred to the appendix.

**Notation.** For $n \in \mathbb{N}$, let $[n] = \{1, ..., n\}$. We generally denote scalars by non-bold letters, vectors by bold lower-case letters and matrices by bold upper-case letters. All vectors $\mathbf{x} \in \mathbb{R}^d \equiv \mathbb{R}^{d \times 1}$ are column vectors, while $\mathbf{x}^\mathsf{T} \in \mathbb{R}^{1 \times d}$ represents its transpose, a row vector. For a matrix $\mathbf{X} \in \mathbb{R}^{d_1 \times d_2}$, let $\mathbf{X}^\mathsf{T} \in \mathbb{R}^{d_2 \times d_1}$ be its transpose. $\mathbf{I}_r$ denotes the identity matrix of size $r$. For $\mathbf{X} \in \mathbb{R}^{d \times d}$, let $\mathrm{trace}(\mathbf{X}) = \sum_{i=1}^{d} \mathbf{X}_{i,i}$. We denote by $\mathbf{A} \succ 0$ and $\mathbf{A} \succeq 0$ if the matrix $\mathbf{A}$ is positive definite and positive semi-definite respectively.

## 2 BACKGROUND

**Algorithmic fairness.** In the last decade, fairness in ML has established itself as a very active area of research which tries to ensure that predictive algorithms are not discriminatory towards any individual or subgroup of population, based on demographic characteristics such as race, gender, disabilities, sexual orientation, or political affiliation (Barocas et al., 2018). Although fair ML is a relatively new area of concern, the growing amount of evidence of discrimination found in increasingly varied fields, has driven the development of several approaches to this problem. We refer to Wang et al. (2022) for a brief review on algorithmic fairness.

In general, the different formalizations of the concept of fairness in the existing literature can be broadly classified into individual and group fairness. Let $\mathbf{X} \in \mathcal{X}$, $\mathbf{S} \in \mathcal{S}$ and $\mathbf{Y} \in \mathcal{Y}$ be the non-sensitive input, the sensitive attribute and the ground truth target variable, respectively. Group (or statistical) fairness emphasizes an equal treatment of individuals with respect to the sensitive attributes $S$, which can be expressed through a measure of statistical independence between the variables involved. In particular, the two main notions along this line are Demographic Parity (DP) (Kamiran & Calders, 2012) and Equality of Odds (EO) (Hardt et al., 2016). The measure DP requires that sensitive attributes should not influence the algorithm's outcome, that is $\hat{Y} \perp\!\!\!\perp S$; while for EO such independence is conditional to the ground-truth, that is $\hat{Y} \perp\!\!\!\perp S|Y$. In the particular setting of binary classification, $\mathcal{Y} = \{0, 1\}$, a classifier $c : \mathbb{R}^d \rightarrow \{0, 1\}$ is said to be DP-fair, with respect to the joint distribution of $(X, S)$, if $P(c(X) = 1|S = s) = P(c(X) = 1)$. On the other hand, $c$ is EO-fair, with respect to $(X, S)$, if $P(c(X) = 1|S = s, Y = y) = P(c(X) = 1|Y = y)$, for $s, y = 0, 1$. A relaxed version of EO has been also proposed as Equality of Opportunity (Hardt et al., 2016) and requires only the equality in TPR, namely $P(c(X) = 1|S = 0, Y = 1) = P(c(X) = 1|S = 1, Y = 1)$. On the other hand, individual fairness (Dwork et al., 2012) examines individual algorithms' predictions and ensures that when two individuals are similar with respect to a specific task, they are classified similarly. However, it is defined in terms of certain similarity metric for the prediction task at hand which is generally difficult to obtain. Individual fairness is close to the notion of counterfactual fairness which specifies the notion of closeness between individual with a causal framework as shown in Kusner et al. (2017); Lara et al. (2024).

Regardless of the notion of fairness, methods for fair forecasting can be divided into *(i) pre-processing* the input data from which the algorithms learns in order to remove sensitive dependencies (Kamiran & Calders, 2012; Calmon et al., 2017; Gordaliza et al., 2019); *(ii) in-processing* by incorporating a fairness constraint or penalty in the algorithm's learning objective function (Zafar et al., 2017; Donini et al., 2018; Risser et al., 2022); and *post-processing*, which modifies the predictions given by the algorithm (Wei et al., 2021).

**Fair Representation Learning.** The field of Fair Representation Learning (FRL) focuses on learning data representations from which any information about the protected group membership has been

removed, while simultaneously retaining as much information related to other features as possible. Hence, any ML model trained on the new representation should not not be able to discriminate based on the demographic information, achieving fair outcomes.

The goal of fair representation is to learn a fair feature representation $r : \mathcal{X} \to \mathcal{X}'$ such that the information shared between $r(\mathbf{X})$ and some sensitive attribute $S \in \mathcal{S}$ is minimal. This is founded on the data processing inequality, a concept from information theory which states that the models' prediction can not have any more information about $S$ than its input or hidden states (Beaudry & Renner, 2012). Hence, the idea is to map the inputs $\mathbf{X}$ to $r(\mathbf{X})$ and use this feature space as input, thus ensuring the certain definition of fairness is achieved, inspired by an ethical notion that establishes the way to limit the influence of $S$ on the outcome of an AI system. As causal dependence is a special kind of statistical dependence (Pearl, 2009), the real aim is to learn a map $r : \mathcal{X} \to \mathcal{X}'$ such that $r(\mathbf{X})$ is (approximately) statistically independent with respect to the sensitive attribute $S$, guaranteeing fairness of any model trained on top of this new representation.

FRL has been initially considered by Zemel et al. (2013) where they propose to learn a representation that is a probability distribution over clusters where learning the cluster of a sample does not give any information about the sensitive attribute $S$. Since then, a variety of methods have been put forward in the recent literature. A popular approach to address this challenge is the variational auto-encoder (VAE) (Gupta et al., 2021; Louizos et al., 2015) which aim to minimize the information encoded in them. Other methods (Edwards & Storkey, 2016; Madras et al., 2018; Xie et al., 2017; Liao et al., 2019) obtain learning representations formulating the problem as an adversarial game, learning an encoder and an adversary. In contrast to the adversarial training scheme, Olfat & Aswani (2019) introduced the concept of fair PCA, aiming to ensure that no linear classifier can predict demographic information from the projected data. This approach has been further extended by Kleindessner et al. (2023); Lee et al. (2022a). Additionally, another notion of fair PCA was proposed by Samadi et al. (2018) which seeks to balance the excess reconstruction error across different demographic groups. This is extended by approaches such as Pelegrina et al. (2021); Kamani et al. (2022).

In this work, we propose to introduce fairness constraints for PLS decomposition. Actually, in a supervised setting, PLS enables to build features which are more accurate than PCA components since they are directly related to the target to be forecast. Hence we propose to extend the PLS feature construction extraction with a fairness constraint, achieving a representation that is both fair but also enables to obtain accurate predictions. This work extends the previous vanilla method described in Champion et al. (2023) and applied to a medical dataset that projects a posteriori the components onto the less biased components characterized by weaker correlations with the biased variable.

In this paper we provide a feasible way to impose fairness constraint on PLS components. Hence we provide representations that both enable to achieve a good forecast accuracy with few components while reducing unwanted biases. Most FRL methods are unsupervised, and existing supervised techniques do not account for situations where the number of samples is smaller than the number of features. Our proposal, Fair PLS, addresses this gap.

## 3 FAIR PARTIAL LEAST SQUARES

Let, as before, $\mathbf{X} \in \mathcal{X}$, $\mathbf{S} \in \mathcal{S}$ and $\mathbf{Y} \in \mathcal{Y}$ be the non-sensitive input, the sensitive attribute and the ground truth target variable, respectively. The samples are drawn from a distribution $\mathbb{P}$ over $\mathcal{X} \times \mathcal{S} \times \mathcal{Y}$, where $\mathcal{X} \subset \mathbb{R}^d$ is the set of possible (non-sensitive) inputs, $\mathcal{Y} \subset \mathbb{R}^m$ is the set of possible labels and $\mathcal{S} \subset \mathbb{R}^{n_s}$ is the set of possible sensitive variable values. In the context of supervised learning, a decision rule, denoted by $f : \mathcal{X} \to \mathcal{Y}$, is built to perform a specific prediction task from a set of labeled samples $\mathcal{D} = \{\mathbf{x}_i, \mathbf{s}_i, \mathbf{y}_i\}_{i=1}^n$. We represent the dataset of $n$ points $\mathbf{x}_1, \mathbf{x}_2, ..., \mathbf{x}_n \in \mathbb{R}^d$ as a matrix $\mathbf{X} \in \mathbb{R}^{n \times d}$, where the $i$-th row is equal to $\mathbf{x}_i$. Without loss of generality, we suppose that $\mathbf{X} \in \mathbb{R}^{n \times d}$ is the original centred $d$ variables of $n$ observations and $\mathbf{Y} \in \mathbb{R}^{n \times m}$ be the centred target.

In this section, we begin by reviewing the Partial Least Squares (PLS) technique and then we introduce the first theoretical formulation of Fair Partial Least Squares. Our approach involves incorporating a regularization term into the PLS objective.

## 3.1 PARTIAL LEAST SQUARES

The Partial Least Squares (PLS - *a.k.a.* projection on latent structures) approach is a supervised dimension reduction technique which generates orthogonal vectors, also referred to as latent vectors or components, by maximising the covariance between different sets of variables (Höskuldsson, 1988; Rosipal & Krämer, 2006; Abdi, 2010). In detail, PLS aims to decompose the zero-mean matrix $\mathbf{X} = \mathbf{T}\mathbf{P}^\mathsf{T}$ as a product of $k \in [d]$ latent vectors (columns of $\mathbf{T} \in \mathbb{R}^{n \times k}$) and a matrix of weights ($\mathbf{P} \in \mathbb{R}^{d \times k}$), with the constraint that these components explain as much as possible of the covariance between $\mathbf{X}$ and $\mathbf{Y}$. In other words, PLS approach attempts to find directions that help explain both the response $\mathbf{Y}$ and the predictors $\mathbf{X}$. Indeed, any collection of orthogonal vectors that span the column space of $\mathbf{X}$ could be used as the latent vectors. Consequently, to achieve decorrelated components with maximum correlation with $\mathbf{Y}$, additional conditions on the matrices $\mathbf{P}$ and $\mathbf{Q}$ will be required. Specifically, the PLS method finds two sets of weights vectors denoted as $(\mathbf{w}_1, ..., \mathbf{w}_d)$ and $(\mathbf{c}_1, ..., \mathbf{c}_d)$ such that the linear combination of the columns of $\mathbf{X}$ and $\mathbf{Y}$ have maximum covariance. The first pair of vectors $\mathbf{w}$ and $\mathbf{c}$ verify the following optimization problem:

$$Cov(\mathbf{t}, \mathbf{u}) = Cov(\mathbf{X}\mathbf{w}, \mathbf{Y}\mathbf{c}) = \max_{||\mathbf{p}||=||\mathbf{q}||=1} Cov(\mathbf{X}\mathbf{p}, \mathbf{Y}\mathbf{q}), \tag{1}$$

where $Cov(\mathbf{t}, \mathbf{u}) = \frac{\mathbf{t}^\mathsf{T}\mathbf{u}}{n}$ denotes the sample covariance between the score vectors. Once the first latent vector is found, the PLS method undergoes a series of iterations, obtaining the $k \in [d]$ weights vectors such that at each iteration $h$, the vector $\mathbf{w}_h$ is orthogonal to all preceding weight vectors $(\mathbf{w}_1, ..., \mathbf{w}_{h-1})$, namely, $\forall l \in [h-1] : \mathbf{t}_h \perp \mathbf{t}_l$. Without loss of generality, we assume that the dependent variables are just one $\mathbf{Y} = \mathbf{y}$, then $\mathbf{c} = \mathbf{1}$ and $\mathbf{u} = \mathbf{y}$. What this means is that the columns of the weight matrix $\mathbf{W}$ are defined such that the squared sample covariance between the latent components and $\mathbf{Y}$ is maximal, given that these latent components are empirically uncorrelated with each other. Moreover, the vectors $(\mathbf{w}_1, \cdots, \mathbf{w}_k)$ are constrained to have a unit length. To sum up, the weights vectors verify the following optimization problem:

$$\forall h \in [k], \quad \mathbf{w}_h = \arg\max_{\mathbf{w} \in \mathcal{W}_h} \quad Cov(\mathbf{X}\mathbf{w}, \mathbf{Y}), \tag{2}$$

where $\mathcal{W}_h = \{\mathbf{w} \in \mathbb{R}^d \mid \mathbf{w}^\mathsf{T}\mathbf{w} = 1, \quad \mathbf{w}^\mathsf{T}\mathbf{X}^\mathsf{T}\mathbf{X}\mathbf{w}_l = 0 \quad \forall l \in [h-1]\}$ and the latent vector are defined as $\mathbf{t}_h = \mathbf{X}\mathbf{w}_h$.

The Nonlinear Iterative Partial Least Squares (NIPALS) was introduced by Wold (1975) as an iterative algorithm for computing the matrices $\mathbf{W}$ and $\mathbf{T}$. The pseudo code can be found in Appendix A. When considering the relationship between vectors at step $h$ and their corresponding vectors at step $h-1$ for a specific dimension, the equations reveal that the NIPALS algorithm performs similarly to the power method used for determining the largest eigenvalue of a matrix. Hence, PLS is closely related to the eigen and singular value decomposition (refer to Abdi (2006) for an introduction to these notions). At convergence of the algorithm, the vector $\mathbf{w}$ satisfies $\mathbf{X}^\mathsf{T}\mathbf{Y}\mathbf{Y}^\mathsf{T}\mathbf{X}\mathbf{w} = \lambda\mathbf{w}$, indicating that the weight vector $\mathbf{w}$ is the first eigenvector of the symmetric positive semi-definite matrix $\mathbf{X}^\mathsf{T}\mathbf{Y}\mathbf{Y}^\mathsf{T}\mathbf{X}$, with $\lambda$ the maximum eigenvalue.

As a consequence, the problem of finding the vectors $\mathbf{w}$ and $\mathbf{c}$ such that the components $\mathbf{t}$ and $\mathbf{u}$ are the ones with maximal covariance among all components in $\mathbf{X}$ and $\mathbf{Y}$ space respectively, is equivalent to the problem of computing the singular vectors of the singular value decomposition (SVD) of the matrix $\mathbf{A} = \mathbf{X}^\mathsf{T}\mathbf{Y}$. This is, the weight vector $\mathbf{w}_1$ is the first left singular vector of the matrix $\mathbf{A}$. The $\mathbf{A}$ can be decompose using a Singular Value Decomposition (SVD) as: $\mathbf{A} = \mathbf{X}^\mathsf{T}\mathbf{Y} = \mathbf{F}\mathbf{\Sigma}\mathbf{G}^\mathsf{T}$, where $\mathbf{F} \in \mathbb{R}^{d \times d}$ contains the left singular vectors, $\mathbf{\Sigma} \in \mathbb{R}^{d \times m}$ is a diagonal matrix with the singular values as diagonal elements, and $\mathbf{G} \in \mathbb{R}^{m \times m}$ contains the right singular vectors. Note that $\mathbf{F}$ and $\mathbf{G}$ are orthogonal matrices, which means that $\mathbf{F}^\mathsf{T}\mathbf{F} = \mathbf{I}_d$ and $\mathbf{G}^\mathsf{T}\mathbf{G} = \mathbf{I}_m$. The square of the largest singular value $\sigma_1$ is in fact the maximum of equation 2 when $\mathbf{p} = \mathbf{f}_1$ and $\mathbf{q} = \mathbf{g}_1$. The vector $\mathbf{F}^\mathsf{T}\mathbf{x}_i \in \mathbb{R}^k$ is the projection of $\mathbf{x}_i$ onto the subspace spanned by the columns of $\mathbf{F}$, viewed as a point in the lower-dimensional space $\mathbb{R}^k$. This solution gives the maximum value $\sum_{i=1}^d \sigma_i$. Höskuldsson (1988) proved that PLS method is based on the fact that the largest singular value at step $h+1$ is larger that the second largest singular value at step $h$.

## 3.2 OUR FORMULATION OF FAIR PLS

In Fair PLS, we aim to learn a projection of the data matrix $\mathbf{X}$ onto a $k$-dimensional subspace $r_\eta(\mathbf{X})$, dependent on the target $Y$ to be forecast, but such that the covariance dependence measure between

the new data representation and the demographic attribute $S$ is minimal, according to parameter $\eta > 0$. Hence, the objective is to learn a map $r_\eta : \mathcal{X} \to \mathcal{X}'$ such that $r_\eta(\mathbf{X})$, at the same time, enables to estimate accurately the parameter of interest $Y$, but is also statistically independent with respect to the sensitive attribute $S$, ensuring fairness of any model trained on this representation. $\eta$ denotes here the parameter that balances the trade-off between the information contained by the representation related to forecast $Y$, and its unbiasedness with respect to the sensitive attribute $S$.

We formulate the Fair PLS (FPLS) approach as the computation of a matrix of weights $\mathbf{W} \in \mathbb{R}^{d \times k}$ where the column $\mathbf{w}_h = [w_{1,h}, \cdots, w_{d,h}]^\mathsf{T}$ represents the solution, for $h \in [k]$, of the optimization problem equation 2 restricting to vectors such that the projections $\mathbf{w}^\mathsf{T}\mathbf{x}_i$ and the sensitive $s_i$ are statistically independent $\forall i \in [n]$. We modify the initial definition of PLS by computing the quadratic covariance. Actually, our method aims at constructing the linear combination of the most correlated components, regardless of the sign. According to Belrose et al. (2023), the fact that every linear classifier exhibits demographic parity with respect to $S$ when evaluated on $\mathbf{X}$ is equivalent to the condition that every component of $\mathbf{X}$ has zero covariance with every component of $S$. In summary, Fair PLS is formulated as:

$$\arg\max_{\mathbf{w} \in \mathcal{W}'_h} \quad Cov^2(\mathbf{X}\mathbf{w}, \mathbf{Y}), \qquad \text{where}$$

$$\mathcal{W}'_h = \{ \mathbf{w} \in \mathbb{R}^d \mid \mathbf{w}^\mathsf{T}\mathbf{w} = 1, \quad \mathbf{w}^\mathsf{T}\mathbf{X}^\mathsf{T}\mathbf{X}\mathbf{w}_l = 0 \quad \forall l \in [h-1] \text{ and}$$

$$\forall i \in [n] \quad Cov^2(\mathbf{w}^\mathsf{T}\mathbf{x}_i, s_i) = 0 \}. \tag{3}$$

The independence criteria between the projections and the sensitive variable, $Cov^2(\mathbf{X}\mathbf{w}, S) = 0$, is added to the optimization problem of the standard PLS technique as a regularization term. Hence, the following general objective function has to be optimise:

$$\forall h \in [k] \quad \mathbf{w}_h = \arg\max_{\mathbf{w} \in \mathcal{W}_h} \quad (\mathbf{C}^2_{\mathbf{X}\mathbf{w}, \mathbf{Y}} - \eta\, \mathbf{C}^2_{\mathbf{X}\mathbf{w}, S}) \equiv$$

$$\forall h \in [k] \quad \mathbf{w}_h = \arg\max_{\mathbf{w} \in \mathcal{W}_h} \quad \left( \frac{1}{n^2}\, \mathbf{w}^\mathsf{T}\mathbf{X}^\mathsf{T}\mathbf{Y}\mathbf{Y}^\mathsf{T}\mathbf{X}\mathbf{w} - \eta\, \frac{1}{n^2}\, \mathbf{w}^\mathsf{T}\mathbf{X}^\mathsf{T}SS^\mathsf{T}\mathbf{X}\mathbf{w} \right), \tag{4}$$

where $\eta > 0$ is the regularization parameter, $\mathbf{C}_{\mathbf{A},\mathbf{B}} = \frac{1}{n}\mathbf{A}^\mathsf{T}\mathbf{B}$ is the empirical cross covariance matrix between $\mathbf{A}$ and $\mathbf{B}$ and $\mathcal{W}_h = \{ \mathbf{w} \in \mathbb{R}^d \mid \mathbf{w}^\mathsf{T}\mathbf{w} = 1, \quad \mathbf{w}^\mathsf{T}\mathbf{X}^\mathsf{T}\mathbf{X}\mathbf{w}_l = 0 \quad \forall l \in [h-1] \}$. This problem is solved in an efficient manner with the Gradient Descent algorithm (Shalev-Shwartz & Ben-David, 2014). Then, at each iteration, we take a step in the direction of the negative of the gradient at the current point. That is, the update step is: $\mathbf{w}^{t+1} = \mathbf{w}^t - \varepsilon\, \frac{\partial g_{FPLS}(\mathbf{w}^t)}{\partial \mathbf{w}}$, where the function to optimize is $g_{FPLS}(\mathbf{w}) = \frac{1}{n^2}\, \mathbf{w}^\mathsf{T}(\mathbf{X}^\mathsf{T}\mathbf{Y}\mathbf{Y}^\mathsf{T}\mathbf{X} - \eta\mathbf{X}^\mathsf{T}SS^\mathsf{T}\mathbf{X})\mathbf{w}$, the respective gradient is $\frac{\partial g_{FPLS}}{\partial \mathbf{w}} = \frac{2}{n^2}\, (\mathbf{X}^\mathsf{T}\mathbf{Y}\mathbf{Y}^\mathsf{T}\mathbf{X} - \eta\mathbf{X}^\mathsf{T}SS^\mathsf{T}\mathbf{X})\mathbf{w}$, and $\varepsilon > 0$ is the learning rate.

---

**Algorithm 1:** Fair PLS algorithm

---

**Input:** $d$ independent variables stored in a centred matrix $\mathbf{X} \in \mathbb{R}^{n \times d}$ and $m$ dependent variables stored in a centred matrix $\mathbf{Y} \in \mathbb{R}^{n \times m}$; sensitive centred variable $S$; $\eta$ parameter; $k$ number of components.

**Output:** $\mathbf{W}$, $\mathbf{T}$.

Set $\mathbf{X}_1 = \mathbf{X}$ and $\mathbf{Y}_1 = \mathbf{Y}$;

**for** $h \in [k]$ **do**

    Compute the weights $\mathbf{w}_h \in \mathbb{R}^d$ as the maximum of the function

    $f_{FPLS}(\mathbf{w}) = \frac{1}{n^2}\, \mathbf{w}^\mathsf{T}\mathbf{X}_h^\mathsf{T}\mathbf{Y}_h\mathbf{Y}_h^\mathsf{T}\mathbf{X}_h\mathbf{w} - \eta\frac{1}{n^2}\, \mathbf{w}^\mathsf{T}\mathbf{X}_h^\mathsf{T}SS^\mathsf{T}\mathbf{X}_h\mathbf{w}$;

    Scale them to be of length one;

    Project $\mathbf{X}_h$ on the singular vectors in order to obtain the scores $\mathbf{t}_h = \mathbf{X}_h\mathbf{w}_h$;

    Compute the loadings $\boldsymbol{\gamma}_h \in \mathbb{R}^d$ such that the matrix of 1-rank $\boldsymbol{\kappa}_h\boldsymbol{\gamma}_h^\mathsf{T}$ is as close as possible

    to $\mathbf{X}_h$;

    Compute residual matrices: $\mathbf{X}_{h+1} = \mathbf{X}_h - \boldsymbol{\kappa}_h\boldsymbol{\gamma}_h^\mathsf{T}$;

**end**

Store the vectors $\mathbf{w}$, $\mathbf{t}$ in the corresponding matrices;

---

Therefore, Fair PLS finds a best approximating projection such that the projected data is statistically independent from the sensitive attribute. The parameter $\eta$ can be interpreted as the trade-off between

fairness and utility. The algorithm that implements this approach is detailed below, and consists of approximating $\mathbf{X}$ as a sum of 1-rank matrices $\mathbf{X} = \mathbf{T}\mathbf{W}^\top$, where $\mathbf{T} \in \mathbb{R}^{n \times k}$ contains the scores in its columns and $\mathbf{W}$ contains the weights in its columns.

**Why cannot Fair PLS be formulated in closed form?**   It is important to note that adapting the Partial Least Squares methodology to achieve fairness as a trade-off is challenging due to the inherent complexity of the PLS method. Contrary to PCA analysis for which a closed form may be found, computing the PLS components is not a direct method. We refer for instance to Blazere et al. (2015) or Löfstedt (2024) and references therein. When modifying the loss with the fairness penalty makes the computation even less tractable. Specifically, if we denote as $\sigma_{min,Y}$ the minimum eigenvalue of the matrix $\mathbf{Y}\mathbf{Y}^\top$ and $\sigma_{max,S}$ the maximum eigenvalue of the matrix $SS^\top$. The Fair PLS weights $\{\mathbf{w}_h\}_{h=1}^k$ are the eigenvectors of the certain matrix if $\eta \leq \sigma_{min,Y}/\sigma_{max,S}$. Moreover the matrix whose eigenvectors are the Fair PLS weights is $\mathbf{X}^\top \mathbf{M}\mathbf{M}^\top \mathbf{X}$, with $\mathbf{B} = \mathbf{Y}\mathbf{Y}^\top - \eta SS^\top = \mathbf{Q}^\top \mathbf{D}\mathbf{Q}$ and $\mathbf{M} = \mathbf{Q}^\top \mathbf{D}^{1/2}$ .

**Motivation for Fair PLS**   The motivation behind the idea of Fair Representation Learning by a PLS-based approach is interpretability. Yet our aim is to provide a method that enables us to recover a linear transformation of the data to promote explainability of the components. Hence, the PLS method was a suitable way to achieve interpretability of the new components yet enabling forecasting. Fair PLS allows us to learn a new representation that not only is lower in dimension but also a trade-off between fairness and utility performance. For instance, for the COMPAS dataset, if we obtain the most relevant features of the learned components, we discover that if $\eta = 0.0$ (i.e. standard PLS), these are: event, decile score, juv misd count, race and decile score; while for $\eta = 1.0$ they are: event, age, juv other count, juv misd count and priors count. Hence, the sensitive variable does not impact the Fair PLS components.

## 4 EXTENSIONS

### 4.1 KERNELIZING FAIR PLS

Let us now extend our Fair PLS approach (Section 3.2) to the non-linear version of PLS by means of reproducing kernels (Rosipal & Trejo, 2002). In this section, we formulate Kernel Fair Partial Least Squares by adding the Hilbert Schmidt independence criterion (Tan et al., 2020; Fukumizu et al., 2007) as the fairness regularization term in the standard PLS formulation in equation 2. The proposed Kernel Fair PLS is based on a fair adaptation of the NIPALS procedure to iteratively estimate the desired components which are not linearly related to the input variables. Furthermore, this will allow to use multiple sensitive attributes simultaneously. To this end, Kernel Fair PLS is a generalization of Fair PLS to feature spaces of arbitrary large dimensionality. We additionally provide the pseudo code of kernelized Fair PLS in Appendix A.

To do this, we assume a nonlinear transformation of the input variables $\mathbf{X} \in \mathbb{R}^{n \times d_X}$ and $\mathbf{S} \in \mathbb{R}^{n \times d_S}$ into separable feature reproducing kernel Hilbert spaces (RKHSs) $(\mathcal{H}_{K_X}, \langle \cdot, \cdot \rangle_{K_X})$ and $(\mathcal{H}_{K_S}, \langle \cdot, \cdot \rangle_{K_S})$, respectively. Recall that in our proposal $d_S \geq 1$ admits more than one sensitive variable. The corresponding mapping functions are defined as $\phi : \mathbf{x}_i \in \mathbb{R}^{d_X} \mapsto \phi(\mathbf{x}_i) \in \mathcal{H}_{K_X}$ and $\psi : \mathbf{s}_i \in \mathbb{R}^{d_S} \mapsto \psi(\mathbf{s}_i) \in \mathcal{H}_{K_S}$, respectively. This yields to the matrices $\mathbf{\Phi}$ and $\mathbf{\Psi}$ where the row $i, 1 \leq i \leq n$ denotes the vectors $\phi(\mathbf{x}_i)$ and $\psi(\mathbf{z}_i)$ respectively. Hence, the corresponding reproducing kernel functions can be written in the form of $K_{\mathbf{X}}(\mathbf{x}_i, \mathbf{x}_j) = \langle \phi(\mathbf{x}_i), \phi(\mathbf{x}_j) \rangle_{K_X} = \phi(\mathbf{x}_i)^\top \phi(\mathbf{x}_j)$ and $K_{\mathbf{S}}(\mathbf{s}_i, \mathbf{s}_j) = \langle \psi(\mathbf{s}_i), \psi(\mathbf{s}_j) \rangle_{K_S} = \psi(\mathbf{s}_i)^\top \psi(\mathbf{s}_j)$, which correspond to the Euclidean dot product in their respective Hilbert spaces. Applying the so-called "kernel trick" (i.e. $\mathbf{\Phi}\mathbf{\Phi}^\top \in \mathbb{R}^{n \times n}$ represents the kernel Gram matrix $\mathbf{K}_{\mathbf{X}}$ of the cross dot products between all mapped input data points), we rewrite equation 4 in terms of the kernel matrix $\mathbf{K}_{\mathbf{X}}$ and $\mathbf{K}_{\mathbf{S}}$. Recall that in Fair PLS approach (Section 3.2), we measured the independence with respect to the sensitive attribute through the cross covariance operator between the components and the protected feature. When kernelising this method, to measure independence we will use the Hilbert Schmidt Independence Criterion (HSIC) introduced in Gretton et al. (2007). To this end, let us provide the functional analytic background necessary to describe cross-covariance operators between RKHSs and introduce the HSIC.

**Definition 1** Cross covariance operator and Hilbert-Schmidt Independence Criterion

*We assume that $(X, \Gamma)$ and $(S, \Lambda)$ are settled up with probability measures $p_x$ and $p_s$ respectively ($\Gamma$ being the Borel sets on $X$, and $\Lambda$ the Borel sets on $S$). Following Yamanishi et al. (2004) and Gretton et al. (2005) the cross-covariance operator associated with the joint measure $P_{xy}$ on $(\mathcal{X} \times \mathcal{Y}, \Gamma \times \Lambda)$ is a linear operator $\mathcal{C}_{XS} : \mathcal{H}_{K_X} \rightarrow \mathcal{H}_{K_S}$ defined as:*

$$\mathcal{C}_{XS} := \mathbb{E}_{XS}[(\phi(X) - \mu_X) \otimes (\psi(S) - \mu_S)] = \mathbb{E}_{XS}[\phi(X) \otimes \psi(S)] - \mu_X \otimes \mu_S. \tag{5}$$

*Given a sample $\{(\boldsymbol{x}_1, \boldsymbol{s}_1), \cdots, (\boldsymbol{x}_n, \boldsymbol{s}_n)\}$ the empirical cross-covariance operator $\mathbf{C}_{X,S} : \mathcal{H}_K \rightarrow \mathcal{H}_{K_S}$ is defined as:*

$$\mathbf{C}_{X,S} := \frac{1}{n} \sum_{i=1}^{n} [\phi(\boldsymbol{x}_i) \otimes \psi(\boldsymbol{s}_i)] - \hat{\mu}_{\boldsymbol{x}} \otimes \hat{\mu}_{\boldsymbol{s}}, \tag{6}$$

*where $\hat{\mu}_{\boldsymbol{x}} = \frac{1}{n} \sum_{i=1}^{n} \phi(\boldsymbol{x}_i)$ and $\hat{\mu}_{\boldsymbol{s}} = \frac{1}{n} \sum_{i=1}^{n} \psi(\boldsymbol{s}_i)$.*

*The Hilbert-Schmidt Independence Criterion (HSIC) is defined as the squared HS-norm of the cross-covariance operator $\mathcal{C}_{XS}$. Then $HSIC(P_{XS}, \mathcal{H}_K, \mathcal{H}_{K_S}) := \|\mathcal{C}_{XS}\|_{HS}^2$.*

To sum up, we formulate Kernel Fair PLS as an optimization problem rewritten in terms of the Kernel matrices, where fairness is incorporated as a regularization term detecting statistical independence through the $HSIC(P_{\phi(\mathbf{X})\mathbf{w}, \psi(\mathbf{S})}, \mathcal{H}_K, \mathcal{H}_{K_S})$ operator. By the Representer's Theorem, the weight can be expressed as $\mathbf{w} = \boldsymbol{\Phi}^{\mathsf{T}} \alpha$ (Rosipal & Trejo, 2002). Hence, the Kernel Fair PLS (KFPLS) is:

$$\forall h \in [k] \quad \mathbf{w}_h = \underset{\mathbf{w} \in \mathcal{W}_h^{Kernel}}{\arg\max} \quad \mathbf{C}_{\phi(\mathbf{X})\mathbf{w}, \mathbf{Y}}^2 - \eta \, \|\mathbf{C}_{\phi(\mathbf{X})\mathbf{w}, \psi(\mathbf{S})}\|_{HS}^2 \equiv$$

$$\forall h \in [k] \quad \alpha_h = \underset{\alpha \in \aleph_h}{\arg\max} \quad \left( \frac{1}{n^2} \operatorname{Tr}(\alpha^{\mathsf{T}} \widetilde{\mathbf{K}}_{\mathbf{X}} \mathbf{Y} \mathbf{Y}^{\mathsf{T}} \widetilde{\mathbf{K}}_{\mathbf{X}} \alpha) - \eta \, \frac{1}{n^2} \operatorname{Tr}(\alpha^{\mathsf{T}} \widetilde{\mathbf{K}}_{\mathbf{X}} \widetilde{\mathbf{K}}_{\mathbf{S}} \widetilde{\mathbf{K}}_{\mathbf{X}} \alpha) \right), \tag{7}$$

where $\eta > 0$ is the regularization parameter and $\mathcal{W}_h^{Kernel} = \{\mathbf{w} \in \mathbb{R}^d \mid \mathbf{w}^{\mathsf{T}}\mathbf{w} = 1, \quad \mathbf{w}^{\mathsf{T}}\boldsymbol{\Phi}^{\mathsf{T}}\boldsymbol{\Phi}\mathbf{w}_l = 0 \quad \forall l \in [h-1]\}$, $\aleph_h = \{\mathbf{a} \in \mathbb{R}^n \mid \mathbf{a}^{\mathsf{T}}\mathbf{K}_{\mathbf{X}}\mathbf{a} = 1, \quad \mathbf{a}^{\mathsf{T}}\mathbf{K}_{\mathbf{X}}\mathbf{K}_{\mathbf{X}}\alpha_l = 0 \quad \forall l \in [h-1]\}$. The Gram matrices for the variables centred in their respective feature spaces are shown by Schölkopf et al. (1998) to be: $\widetilde{\mathbf{K}}_{\mathbf{X}} = \mathbf{H}\mathbf{K}_{\mathbf{X}}\mathbf{H}$ and $\widetilde{\mathbf{K}}_{\mathbf{S}} = \mathbf{H}\mathbf{K}_{\mathbf{S}}\mathbf{H}$, where $\mathbf{H} = \mathbf{I}_n - \frac{1}{n}\mathbf{1}_n\mathbf{1}_n^{\mathsf{T}}$, and $\mathbf{1}_n$ is an $n \times 1$ vector of ones. Then, the matrices $\widetilde{\boldsymbol{\Phi}}$ and $\widetilde{\boldsymbol{\Psi}}$ contain the centered data in Hilbert space. In the case where the kernel is $\mathbf{K}(\boldsymbol{a}, \boldsymbol{b}) = \langle \boldsymbol{a}, \boldsymbol{b} \rangle$ we recover the Fair PLS approach.

## 4.2 IMPOSING EQUALITY OF ODDS CONSTRAINT

The aim of Fair PLS, as formulated in Section 3, is to represent the data such that it is independent of the demographic attribute. This approach guarantees that any classifier trained with the new data achieves demographic parity fairness. Yet the methodology we develop can be extended to other notions of global fairness, for instance to Equality of Odds (EO) or its relaxed version Equality of Opportunity. As mentioned before, EO can be mathematically expressed as the conditional independence $\hat{Y} \perp S \mid Y$, in the sense that the forecast error should not depend on the sensitive attribute. If we aim to attain EO, we could apply the methodology of Fair PLS to the input data where we replace $\hat{Y}$ by $\mathbf{X}\mathbf{w}$. In this case, the EO condition will hold for any function which is built with the PLS directions $\mathbf{X}\mathbf{w}$'s. In this case we can define the EO Fair PLS estimator as:

$$\forall h \in [k] \quad \mathbf{w}_h = \underset{\mathbf{w} \in \mathcal{W}_h}{\arg\max} \quad (\mathbf{C}_{\mathbf{X}\mathbf{w}, \mathbf{Y}}^2 - \eta \, \mathbf{C}_{\mathbf{X}\mathbf{w}, S|Y}^2), \tag{8}$$

where $\mathbf{C}_{\mathbf{X}\mathbf{w}, S|Y}$ is the conditional cross covariance which is defined as

$$\mathbf{C}_{\mathbf{X}\mathbf{w}, S|Y} = \mathbf{C}_{\mathbf{X}\mathbf{w}S} - \mathbf{C}_{\mathbf{X}\mathbf{w}Y}\mathbf{C}_{YY}^{-1}\mathbf{C}_{YS}.$$

This idea has been already used in Perez-Suay et al. (2023) to impose fairness as equality of odds for regression. Our method for PLS representation can thus be extended to this setting. The RKHS framework still holds by replacing the constraint on the covariance by the HSIC criterion for the conditional cross covariance operator, following the guidelines in Perez-Suay et al. (2023).

## 4.3 APPLICATION TO LARGE LANGUAGE MODELS

In the context of supervised learning, a decision rule to perform a specific classification task is obtained from a set of labeled samples $\mathcal{X}$. However, in the setting of Large Language Models (LLM), such a decision rule is considered to be $f : \mathcal{Z} \to \mathcal{Y}$, where $\mathbf{z} \in \mathcal{Z}$ represents an input text and $y \in \mathcal{Y}$ denotes their corresponding label. Therefore, the decision rule $f$ can be viewed as a composition of two functions $f = c \circ h$. The first one $h : \mathcal{Z} \to \mathcal{A}$ encompasses all the layers to transform the input data $\mathbf{z} \in \mathcal{Z}$ to a vector $\mathbf{a}$ belonging to the latent space $\mathcal{A}$. The second function $c : \mathcal{A} \to \mathcal{Y}$ involves all the layers to classify the transformed data $h(\mathbf{z}) \in \mathbb{R}^d$. We represent the dataset of $n$ points $\mathbf{a}_1, \mathbf{a}_2, ..., \mathbf{a}_n \in \mathbb{R}^d$ as a matrix $\mathbf{A} = h(\mathbf{Z}) \in \mathbb{R}^{n \times d}$, where the $i$-th row is equals $\mathbf{a}_i$. This matrix is known as CLS-embedding matrix in the encoder transformer model. SVD decomposition is a successful way to understand how the embedding matrix can be factorized into concepts that enable to understand the behavior of the language model. This framework has been recently presented in Jourdan et al. (2023b) and bias analysis in this context is discussed in Jourdan et al. (2023a) for instance. Hence, for SVD, the matrix $\boldsymbol{A}$ is decomposed into $\boldsymbol{A} = U_0 \Sigma_0 V_0^\top$, where $U_0 \in \mathbb{R}^{n \times n}$ and $V_0 \in \mathbb{R}^{d \times d}$ are orthonormal matrices, and $\Sigma_0 \in \mathbb{R}^{n \times d}$ is diagonal. This decomposition reveals the main variability in $\boldsymbol{A}$. By retaining the $r \ll d$ largest singular values in $\Sigma$, we approximate $\boldsymbol{A}$ as: $\boldsymbol{A} \approx \boldsymbol{UW}$, where $\boldsymbol{U} \in \mathbb{R}^{n \times r}$ contains the leading $r$ columns of $U_0$, and $\boldsymbol{W} = \Sigma V^\top \in \mathbb{R}^{r \times d}$ combines the singular values and right singular vectors (see Eckart & Young (1936)). SVD is used to capture the most significant patterns in the data by reducing dimensionality while preserving maximum variance. For our purposes, $\boldsymbol{U}$ is the concept matrix that is needed to be fair. The PLS-based approach is relevant to not lose explainability of the new components due to the linear transformation.

An application of the Fair PLS methodology proposed is to reduce the influence of demographic factors that contribute to the model predictions using an algebraic decomposition of the latent representations of the model into orthogonal dimensions. In other words, we aim to intervene in the latent representations to generate a new fair representation with respect to dimensions that convey bias. This approach is justified due to the orthogonality of the dimensions of the new representation obtained with Fair PLS.

## 5 EXPERIMENTAL RESULTS

In this section, we present a number of experiments [1] conducted on six public datasets to demonstrate the effectiveness of our approach in achieving both fair representation (experiment $\mathcal{A}$) and fair predictions for classification and regression tasks (experiment $\mathcal{B}$). Therefore, in a first phase we checked with these real datasets that the proposed method achieves a good and fair representation of the data. Then, in a second phase, we used such representation for prediction purposes and look at its efficiency in achieving a fairness-accuracy trade-off.

Fairness as DP is usually measured through the so-called Disparate Impact (DI) index, namely $DI(\hat{Y}, S) = P(\hat{Y} = 1|S = 0)/P(\hat{Y} = 1|S = 1)$, which can be empirically estimated as: $\frac{n_{1,0}}{n_{0,0}+n_{1,0}} / \frac{n_{1,1}}{n_{0,1}+n_{1,1}}$, where $n_{i,j}$ is the number of observations such that $Y = i, S = j$. Additionally, Confidence intervals (with $95\%$ confidence) were computed using the method described in Besse et al. (2022). All tables and figures presented in this section, as well as in the supplementary appendix to this section, contain average results (together with standard deviations) over 3 random splits into train and test data for ($\mathcal{A}$) and 7 random splits for ($\mathcal{B}$), respectively. Further information regarding the datasets and implementation details analyzing runtime performance and comparisons with state-of-the-art methods, can be found in Appendix B.

**(Experiment $\mathcal{A}$) Fair Representation.**     The primary goal of our approach is to learn from the original data $\mathbf{X} \in \mathbb{R}^{n \times d}$ a new representation $r_\eta(\mathbf{X}) \in \mathbb{R}^{n \times k}$, in a way that, at the same time, it

> **(Result $\mathcal{A} - 1$)** is useful for target prediction;
>
> **(Result $\mathcal{A} - 2$)** is approximately independent of the sensitive variable; and
>
> **(Result $\mathcal{A} - 3$)** preserves information about the input space.

---

[1]Code available on GitHub repository

In order to check that we effectively achieve all three results, the analyses carried out consist, based on our Fair PLS formulation, of evaluating the behaviour of the covariance between the projections and the target, and between the projections and the sensitive attribute, as the parameter $\eta$ increases for six different datasets.

Evidence is shown in Table 1 and Figure 2, where for all datasets, we make the parameter $\eta$ vary in $[0, 10]$ (see first column). The second column of Table 1 compares in terms of $Cov^2(r_\eta(\mathbf{X}), Y)$ how important the choice of $\eta$ is for building a representation that is balanced ($\mathcal{A} - 1$). Moreover, the third column shows the dependence between the new fair representation and the sensitive variable through $Cov^2(r_\eta(\mathbf{X}), S)$ ($\mathcal{A} - 2$). Both results can also be seen in the blue and orange lines, respectively, in Figure 2. First, notice that in order to achieve a good representation in terms of balance between predictive performance (PLS objective function) and fairness (constraint added to PLS), the parameter $\eta$ should not be much higher than the value 1, which is in fact the fixed parameter for the $Cov^2(r_\eta(\mathbf{X}), Y)$ term in equation 4. Furthermore, if $\eta >> 2$, the new representation lacks of achieving the supervised learning purpose, since the values of $Cov^2(r_\eta(\mathbf{X}), Y)$ decrease considerably. The second experiment related to the representation itself consists of studying the amount of information preserved from the original data, which can be quantified through the reconstruction error $Error(X, r_\eta(\mathbf{X})) = Tr((\mathbf{X} - r_\eta(\mathbf{X}))^\mathsf{T}(\mathbf{X} - r_\eta(\mathbf{X})))$ . The results are displayed in the last column of Table 1 together with Figure 3. The reconstruction error could be interpreted as the variability of the data which we are not able to capture in the lower dimensional space. As the ignored subspace is the orthogonal complement of the principal subspace, then the reconstruction error can be seen as the average squared distance between the original data points and their respective projections onto the principal subspace. For our purposes, an optimal representation is one for which the reconstruction error is small, as is the case with the COMPAS and Communities and Crimes Datasets.

Table 1: The table summarizes the three results: ($\mathcal{A} - 1$) how well the fair representation explains the target variable ($Cov^2(r_\eta(\mathbf{X}), Y)$); ($\mathcal{A} - 2$) how strongly the fair representation is associated with a sensitive variable ($Cov^2(r_\eta(\mathbf{X}), S)$); and ($\mathcal{A} - 3$) how the new representation is close to the original one ($Error(X, r_\eta(\mathbf{X}))$) for different datasets and values of the parameter $\eta$.

| **Dataset** | $\eta$ | $Cov^2(r_\eta(\mathbf{X}), Y)$ | $Cov^2(r_\eta(\mathbf{X}), S)$ | $Error(X, r_\eta(\mathbf{X}))$ |
|---|---|---|---|---|
| | 0.0 | $0.3227 \pm 0.1593$ | $0.1351 \pm 0.0654$ | $0.7891 \pm 0.0045$ |
| Adult Income | 1.0 | $0.29 \pm 0.1488$ | $0.0474 \pm 0.0238$ | $0.794 \pm 0.0034$ |
| | 2.0 | $0.2625 \pm 0.1302$ | $0.021 \pm 0.0107$ | $0.8012 \pm 0.0052$ |
| | 10.0 | $0.2031 \pm 0.1013$ | $0.0016 \pm 0.0005$ | $0.8027 \pm 0.0068$ |
| | 0.0 | $0.0954 \pm 0.0464$ | $0.0455 \pm 0.0234$ | $0.5949 \pm 0.0054$ |
| German Credit | 1.0 | $0.0956 \pm 0.0426$ | $0.0055 \pm 0.0032$ | $0.6094 \pm 0.014$ |
| | 2.0 | $0.0811 \pm 0.0497$ | $0.0078 \pm 0.0046$ | $0.6072 \pm 0.0193$ |
| | 10.0 | $0.0874 \pm 0.0436$ | $0.0062 \pm 0.0021$ | $0.6056 \pm 0.0144$ |
| | 0.0 | $0.0385 \pm 0.0189$ | $0.032 \pm 0.0157$ | $0.2398 \pm 0.0051$ |
| Law School | 1.0 | $0.0129 \pm 0.007$ | $0.0054 \pm 0.0035$ | $0.3915 \pm 0.0125$ |
| | 2.0 | $0.0037 \pm 0.0021$ | $0.0004 \pm 0.0004$ | $0.4603 \pm 0.004$ |
| | 10.0 | $0.0018 \pm 0.001$ | $0.0 \pm 0.0$ | $0.4735 \pm 0.0031$ |
| | 0.0 | $0.0338 \pm 0.0176$ | $0.0073 \pm 0.0038$ | $0.7436 \pm 0.0052$ |
| Diabetes | 1.0 | $0.0326 \pm 0.0174$ | $0.0014 \pm 0.001$ | $0.7399 \pm 0.0025$ |
| | 2.0 | $0.031 \pm 0.0163$ | $0.0006 \pm 0.0005$ | $0.7428 \pm 0.002$ |
| | 10.0 | $0.0289 \pm 0.0147$ | $0.0002 \pm 0.0001$ | $0.7439 \pm 0.0103$ |
| | 0.0 | $0.472 \pm 0.2397$ | $0.0647 \pm 0.0314$ | $0.0953 \pm 0.028$ |
| COMPAS | 1.0 | $0.4518 \pm 0.227$ | $0.0424 \pm 0.0202$ | $0.1521 \pm 0.0109$ |
| | 2.0 | $0.4225 \pm 0.2125$ | $0.028 \pm 0.0145$ | $0.1681 \pm 0.0061$ |
| | 10.0 | $0.3081 \pm 0.1509$ | $0.0061 \pm 0.0029$ | $0.242 \pm 0.0173$ |
| | 0.0 | $0.5563 \pm 0.2898$ | $0.5965 \pm 0.2882$ | $0.1954 \pm 0.011$ |
| Communities and | 1.0 | $0.4948 \pm 0.2653$ | $0.3098 \pm 0.1696$ | $0.209 \pm 0.013$ |
| Crimes | 2.0 | $0.3715 \pm 0.1874$ | $0.153 \pm 0.0922$ | $0.2177 \pm 0.0052$ |
| | 10.0 | $0.1358 \pm 0.0704$ | $0.0098 \pm 0.007$ | $0.2576 \pm 0.0077$ |

**(Experiment $\mathcal{B}$) Fair Predictions.** The final aim of the Fair PLS formulation is to achieve an optimal fair representation so that any ML model trained on $r_\eta(X)$ is fair and has a good predictive performance. For this evaluation, we consider two different settings, classification and regression. Therefore, we used binary target values from five real datasets for the first task (Adult Income, German Credit, Law School, Diabetes and COMPAS datasets) and a positive variable for the second one (Communities and Crimes dataset). The classification results for the Adult and Diabetes datasets are shown and discussed below, while for the rest of datasets, as well as the regression problem, results can be found in Table 4 and Table 5 in Appendix B.

In order to study the trade-off between fairness (DI) and accuracy in Figure 1, several ML models were used. Precisely, logistic regression (LR, in the first column), decision trees (DT, in the second column), and extreme gradient boosting (XGB, in the third column) were trained considering two protected attributes in both cases. Specifically, we applied our method as a pre-processing bias mitigation technique and plot the average values of DI and accuracy obtained from a 7-fold cross-validation, for different values of $\eta \in [0, 2]$. Recall from the previous experiment that these are desirable values for this parameter. In particular, it can be seen that the best trade-off is achieved for $\eta = 1$ in all cases.

Figure 1: ($\mathcal{B}$) Prediction accuracy vs. disparate impact (DI) using various ML models with the new fair representation as input data. Each point represents the average value from a 7-fold cross-validation and the different colors are for the wide range of $\eta$ used to compute the components.

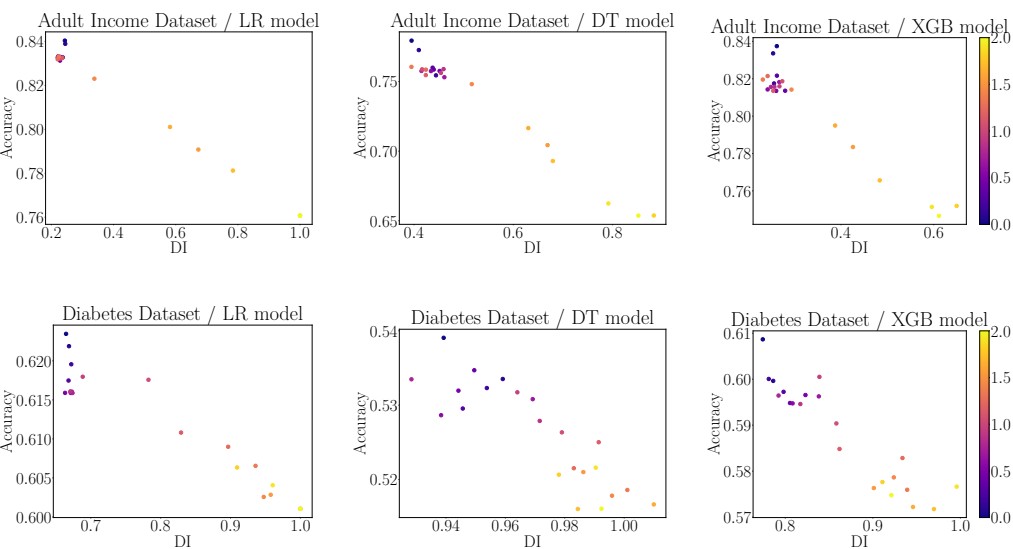

## 6 CONCLUSIONS

We define a Fair Partial Least Squares approach that allows to balance between utility (predictive performance) and fairness (independence of the demographic information) and can be kernelized. Our formulation have the same complexity (algorithmically) as standard Partial Least Squares, or Kernel Partial Least Squares, and have applications on different domains and with different data structures as tabular, image or text embeddings. Furthermore, it can be adapted to the equality of odds paradigm through the use of the conditional cross covariance operator. This poses a robust methodology able to solve different fair scenarios. The experiments demonstrates empirical guarantees of fairness of any model trained on top of the Fair PLS representation and better predictive performance for the same level of fairness when is compared to existing methods for FRL as Fair PCA.

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

## A  APPENDIX TO SECTIONS 3 AND 4.1

---

**Algorithm 2:** Nonlinear Iterative Partial Least Squares (NIPALS): A PLS algorithm

---

**Input:** $d$ independent variables stored in a matrix $\mathbf{X} \in \mathbb{R}^{n \times d}$ and $m$ dependent variables stored in a matrix $\mathbf{Y} \in \mathbb{R}^{n \times m}$.

**Output:** $\mathbf{W}$, $\mathbf{T}$, $\mathbf{C}$, $\mathbf{U}$ and $\mathbf{P}$.

Create two matrices $\mathbf{E} = \mathbf{X}$ and $\mathbf{F} = \mathbf{Y}$;

The matrices $\mathbf{E}$ and $\mathbf{F}$ are column centred and normalized;

Set $\mathbf{u}$ to the first column of $\mathbf{F}$ (could be also initialized with random values);

**while** $\mathbf{E}$ *is not the null matrix* **do**
   **while** $\mathbf{t}$ *not converged* **do**
      $\mathbf{w} = \mathbf{E}^\mathsf{T}\mathbf{u}/(\mathbf{u}^\mathsf{T}\mathbf{u})$;
      Scale $\mathbf{w}$ to be of length one;
      $\mathbf{t} = \mathbf{E}\mathbf{w}$;
      $\mathbf{c} = \mathbf{F}^\mathsf{T}\mathbf{t}/(\mathbf{t}^\mathsf{T}\mathbf{t})$;
      Scale $\mathbf{c}$ to be of length one;
      $\mathbf{u} = \mathbf{F}^\mathsf{T}\mathbf{c}$;
   **end**
   $\mathbf{p} = \mathbf{E}^\mathsf{T}\mathbf{t}/(\mathbf{t}^\mathsf{T}\mathbf{t})$;
   $\mathbf{q} = \mathbf{F}^\mathsf{T}\mathbf{u}/(\mathbf{u}^\mathsf{T}\mathbf{u})$;
   $b = \mathbf{u}^\mathsf{T}\mathbf{t}/(\mathbf{t}^\mathsf{T}\mathbf{t})$;
   Compute the residual matrices: $\mathbf{E} = \mathbf{E} - \mathbf{t}\mathbf{p}^\mathsf{T}$ and $\mathbf{F} = \mathbf{F} - b\mathbf{t}\mathbf{c}^\mathsf{T}$;
   Store the vectors $\mathbf{w}, \mathbf{t}, \mathbf{c}, \mathbf{u}, \mathbf{p}$ in the corresponding matrices;
**end**

---

**Algorithm 3:** Naive Fair PLS

---

**Input:** $d$ independent variables stored in a centred matrix $\mathbf{X} \in \mathbb{R}^{n \times d}$; $m$ dependent variables stored in a centred matrix $\mathbf{Y} \in \mathbb{R}^{n \times m}$; sensitive variable $S$; threshold $\tau$.

**Output:** $\mathbf{T}$ composed of each latent variable $\mathbf{t}_h$ selected.

**for** $h = 1$ *to* $k$ **do**
   Solve $\mathbf{w}_h = \arg\max\limits_{\|\mathbf{w}\|=1} Cov(\mathbf{X}\mathbf{w}, \mathbf{Y})$ ;
   Extract $\mathbf{t}_h = \mathbf{X}\mathbf{w}_h$ ;
   Calculate the correlation ratio $\text{Corr}_h = \eta^2(\mathbf{t}_h, S)$ ;
   **if** $Corr_h < \tau$ **then**
      $\mathbf{t}_h$ is added as a column of $\mathbf{T}$;
   **end**
**end**

---

**Algorithm 4:** Kernel Fair PLS algorithm

---

**Input:** $\mathbf{\Phi} \in \mathbb{R}^{n \times d}$ matrix of mapped input data and $m$ dependent variables stored in a centred matrix $\mathbf{Y} \in \mathbb{R}^{n \times m}$; sensitive mapped data $\mathbf{\Psi}$; $\eta$ parameter; $k$ number of components.

**Output:** $\mathbf{T}$.

Set $\mathbf{K}_{\mathbf{X},1} = \mathbf{K}_{\mathbf{X}}$ and $\mathbf{Y}_1 = \mathbf{Y}$;

Center the matrices $\widetilde{\mathbf{K}}_{\mathbf{X},1} = \mathbf{H}\mathbf{K}_{\mathbf{X},1}\mathbf{H}$ and $\widetilde{\mathbf{K}}_{\mathbf{S},1} = \mathbf{H}\mathbf{K}_{\mathbf{S},1}\mathbf{H}$, where $\mathbf{H} = \mathbf{I}_n - \frac{1}{n}\mathbf{1}_n\mathbf{1}_n^\mathsf{T}$ ;

**for** $h \in [k]$ **do**
   Compute the vector $\alpha_h \in \mathbb{R}^n$ the maximum of the function
   $f_{FKPLS}(\alpha) = \frac{1}{n^2} \text{Tr}(\alpha^\mathsf{T}\widetilde{\mathbf{K}}_{\mathbf{X},h}\mathbf{Y}\mathbf{Y}^\mathsf{T}\widetilde{\mathbf{K}}_{\mathbf{X},h}\alpha) - \eta \frac{1}{n^2} \text{Tr}(\alpha^\mathsf{T}\widetilde{\mathbf{K}}_{\mathbf{X},h}\widetilde{\mathbf{K}}_{\mathbf{S},h}\widetilde{\mathbf{K}}_{\mathbf{X},h}\alpha)$;
   Scale them to be of length one;
   Obtain the scores $\mathbf{t}_h = \widetilde{\mathbf{K}}_{\mathbf{X},h}\alpha_h$ ;
   Compute residual matrices: $\widetilde{\mathbf{K}}_{\mathbf{X},h+1} = \widetilde{\mathbf{K}}_{\mathbf{X},h} - \mathbf{t}_h\mathbf{t}_h^\mathsf{T}\widetilde{\mathbf{K}}_{\mathbf{X},h} - \widetilde{\mathbf{K}}_{\mathbf{X},h}\mathbf{t}_h\mathbf{t}_h^\mathsf{T} + \mathbf{t}_h\mathbf{t}_h^\mathsf{T}\widetilde{\mathbf{K}}_{\mathbf{X},h}\mathbf{t}_h\mathbf{t}_h^\mathsf{T}$;
**end**

Store the vectors $\mathbf{t}$ in the corresponding matrices;

---

# B    APPENDIX TO SECTION 5

## B.1    DETAILS ABOUT DATASETS

**Adult Income dataset**    The Adult Income dataset, available through the UCI repository (Dua & Graff, 2017) provides the results of a census made in 1994 in the United States. Specifically, it contains information about 48842 of individuals, described as values of 14 features: 8 categorical and 6 numeric. The objective of this dataset is to accurately predict whether an individual's annual income is above or below $50,000\$$, taking into account factors such its occupation, marital status, and education.

**German Credit dataset**    The German credit dataset (Hofmann, 1994), comprises records of individuals who hold bank accounts. This dataset serves the purpose of forecasting risk, specifically to assess whether it's advisable to extend credit to an individual. Specifically, it contains information about 1000 individuals, described as values of 21 features: 14 categorical and 7 numerical. The objective of this dataset is to accurately predict the customer's level of risk when granting a credit, taking into account factors such as the status of the existing checking account, credit amount or marital status.

**Law School dataset**    The Law School Admission Council dataset, gathers statistics from 163 US law schools and more than 20,000 students, obtained through a survey across 163 law schools in the United States. This dataset serves the purpose of forecasting the first -year grade from the profile. Specifically, it contains information about 21,791 individuals, described as values of 7 features: 2 categorical and 7 numerical. The objective of this dataset is to accurately predict if an applicant will have a high FYA, taking into account factors such as students entrance exam scores (LSAT), their grade-point average (GPA) collected prior to law school, and their first year average grade (FYA).

**Diabetes dataset**    The Diabetes dataset (Clore et al., 2014), represents ten years (1999-2008) of clinical care at 130 US hospitals and integrated delivery networks. Each row concerns hospital records of patients diagnosed with diabetes, who underwent laboratory, medications, and stayed up to 14 days. This dataset serves the purpose of forecasting if a patient will be readmitted within 30 days of discharge. Specifically, it contains information about 101766 individuals, described as values of 49 features: 36 categorical and 13 numerical. The objective of this dataset is to accurately predict the readmitted $\{< 30, > 30\}$ indicating whether a patient will readmit within 30 days (the positive class is $< 30$), taking into account factors such as weight, gender or the number of lab tests performed during the encounter.

**COMPAS dataset**    The COMPAS dataset, (Angwin et al., 2016), which was released by ProPublica in 2016 is based on the Broward County data (collected from January 2013 to December 2014). This dataset serves the purpose of forecasting recidivism risk scores, specifically to predict if an individual is rearrested within 2 years after the first arrest. Specifically, it contains information about 7214 individuals, described as values of 52 features: 33 categorical and 19 numerical. The objective of this dataset is to accurately predict the COM-PAS recid, taking into account factors such as the risk of recidivism in general, sex or age.

**Communities and Crimes dataset**    The Communities and Crimes dataset (Redmond, 2002), is a small dataset containing the socioeconomic data from 46 states of the United States in 1990 (the US Census). This dataset serves the purpose of forecasting the total number of violent crimes per 100 thousand population. Specifically, it contains information about 1994 individuals, described as values of 127 features: 4 categorical and 123 numerical. The objective of this dataset is to accurately predict the number of violent crimes per 100,000 population (normalized to [0,1]) taking into account factors such as median household income, per capita income or number of kids born to never married.

**Synthetic dataset**    The synthetic dataset contains two groups ($S = 0$ and $S = 1$) with distinct statistical properties. The data includes four quantitative variables (0-3) and three binary variables (4-6), all correlated within each group. We generate 500 samples for each group from two multivariate normal distributions on $\mathbb{R}^7$; with means $(9, 8, 10, 10, 0, 0, 0)$ and $(10, 10, 10, 10, 0, 0, 0)$,

respectively, and different covariance matrix. For females, binary variables 4 and 5 strongly impact variable 0, while variable 6 influences variable 1. In contrast, for males, binary variables have little to no impact on these quantitative variables. The target variable, $Y$, is generated using a weighted combination of these features, with different coefficients for each group. The data is then shuffled, and a binary indicator $S$ is added to distinguish between genders. This setup provides a useful framework for testing biases and statistical analysis.

Table 2: Bias measured in the original datasets. For the datasets whose task is regression (Communities and Crimes) the column of DI is actually the KS value.

| Dataset | Sensitive | Privileged group | Disparate impact | Conf. Interval |
|---|---|---|---|---|
| Adult Income | Gender | Male | 0.3597 | [0.3428 , 0.3765] |
| German Credit | Age | > 25 | 0.7948 | [0.6928 , 0.8968] |
| Law School | Race | White | 0.6713 | [0.6423 , 0.7004] |
| Diabetes | Race | Caucasian | 0.8952 | [0.8758 , 0.9146] |
| COMPAS | Race | Caucasian | 0.8009 | [0.7641 , 0.8378] |
| Communities and Crimes | Race | Not black | 0.129* | - |

## B.2 DETAILS ABOUT IMPLEMENTATION SETUP

**General details.**

- Data pre-processing: the details about how each dataset has been processed can be find in the GitHub repository.

- Data normalization: We normalized the input data to have zero mean and unit variance.

- Dimension of the fair representation: As target dimension we chose $k \in [d]$ with the classical cross validation procedure, where the objective is to find the best trade off of $Cov^2(r(\mathbf{X}), Y) - \eta Cov^2(r(\mathbf{X}), S)$. Notice that the $k$ selected could it be selected differently for each $\eta$.

Table 3: Best number of components $k$ for each dataset in terms of the objective function of the maximization problem equation 4. The value $d$ is the number of features after the preprocessing of the datasets.

| Dataset | $d$ | $k(\eta = 0.0)$ | $k(\eta = 1.0)$ | $k(\eta = 2.0)$ |
|---|---|---|---|---|
| Adult Income | 36 | 5 | 5 | 3 |
| German Credit | 21 | 20 | 7 | 1 |
| Law School | 3 | 3 | 2 | 2 |
| Diabetes | 27 | 20 | 5 | 2 |
| COMPAS | 6 | 6 | 5 | 2 |
| Communities and Crimes | 33 | 7 | 5 | 3 |

**Details about the prediction models.** We propose three different state-of-the-art supervised learning models: Logistic Regression (LR) / Linear Regression (LR), Decision Tree (DT) and Extreme Gradient Boosting (XGB). The selection of these algorithms stems from their ability to encompass distinct modeling approaches, each of them representing different learning paradigms. It is important to note that the aforementioned modeling approaches do not incorporate any algorithmic fairness constraints throughout their modeling process. Consequently, they serve as reference solutions against which bias mitigation techniques can be evaluated and compared. We trained the three prediction models using Scikit-learn and the default specifications for each of them.

## B.3 EXPERIMENTS OF FAIR PARTIAL LEAST SQUARES

Figures 2 and 3 provide the results of the experiment $(\mathcal{A})$ Fair Representations where the aim is to verify that the new representation satisfies the conditions imposed.

Figure 2: (Fair Representation). Comparing the objective functions of the Fair PLS formulation for the representation $r_\eta(\mathbf{X})$. The blue line shows result $(\mathcal{A} - 1)$ while result $(\mathcal{A} - 2)$ is represented by the orange one.

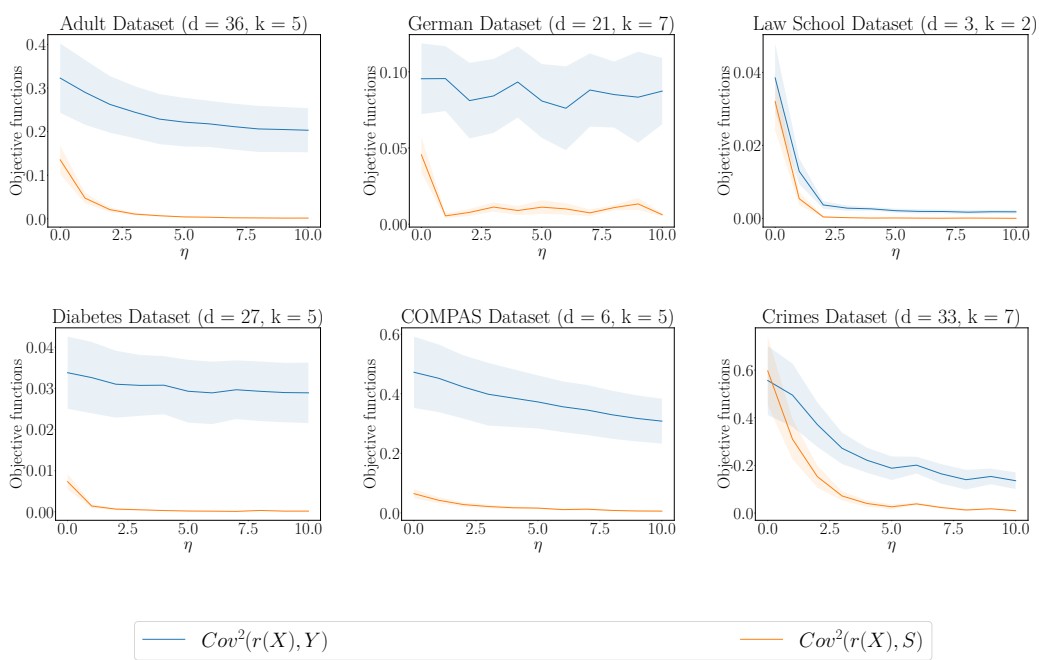

Figure 3: (Fair Representation). The reconstruction error for the new representation $r_\eta(\mathbf{X})$ with respect to the original variables $\mathbf{X}$, showing result $(\mathcal{A} - 3)$.

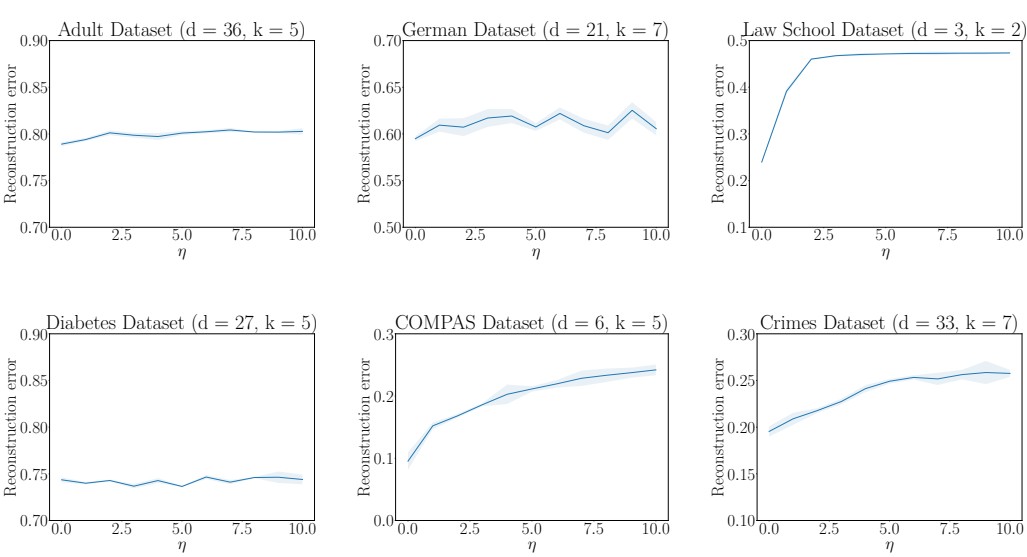

Table 4 and Table 5 provide the results of the experiments for the classification and regression setup respectively. This is, for different values of $\eta$ and diverse datasets, we predict with three ML models and show the accuracy (mean square error) and disparate impact (KS) values for classification (and regression).

Table 4: ($\mathcal{B}$ - Fair Predictions: classification) Results of accuracy and fairness (quantified by the Disparate Impact), for different ML models trained using the Fair-PLS learned representation.

| $\eta$ | Dataset | Model | Disparate impact | Accuracy | $Cov^2(r(X), \hat{Y})$ | $Cov^2(r(X), S)$ |
|---|---|---|---|---|---|---|
| | Adult Income | LR | $0.2433 \pm 0.0255$ | $0.8403 \pm 0.0036$ | $0.2236 \pm 0.0127$ | $0.068 \pm 0.0075$ |
| | | DT | $0.395 \pm 0.026$ | $0.7794 \pm 0.0067$ | $0.1714 \pm 0.013$ | $0.068 \pm 0.0075$ |
| | | XGB | $0.2618 \pm 0.0299$ | $0.8375 \pm 0.0047$ | $0.2231 \pm 0.0129$ | $0.068 \pm 0.0075$ |
| | German Credit | LR | $0.8532 \pm 0.1133$ | $0.716 \pm 0.0315$ | $0.1613 \pm 0.0449$ | $0.0804 \pm 0.0497$ |
| | | DT | $0.8505 \pm 0.1735$ | $0.64 \pm 0.0563$ | $0.1062 \pm 0.0335$ | $0.0804 \pm 0.0497$ |
| | | XGB | $0.8888 \pm 0.1453$ | $0.701 \pm 0.0339$ | $0.1458 \pm 0.0539$ | $0.0804 \pm 0.0497$ |
| 0.0 | Law School | LR | $0.6626 \pm 0.0548$ | $0.9049 \pm 0.0021$ | $0.0099 \pm 0.0053$ | $0.0166 \pm 0.006$ |
| | | DT | $0.6901 \pm 0.0308$ | $0.8475 \pm 0.0056$ | $0.0219 \pm 0.0069$ | $0.0166 \pm 0.006$ |
| | | XGB | $0.719 \pm 0.0487$ | $0.9003 \pm 0.0013$ | $0.0089 \pm 0.0056$ | $0.0166 \pm 0.006$ |
| | Diabetes | LR | $0.6647 \pm 0.0391$ | $0.6235 \pm 0.0024$ | $0.1149 \pm 0.008$ | $0.0047 \pm 0.002$ |
| | | DT | $0.9392 \pm 0.0366$ | $0.5391 \pm 0.006$ | $0.0149 \pm 0.0037$ | $0.0047 \pm 0.002$ |
| | | XGB | $0.7743 \pm 0.0278$ | $0.6087 \pm 0.0047$ | $0.096 \pm 0.0068$ | $0.0047 \pm 0.002$ |
| | COMPAS | LR | $0.8083 \pm 0.0282$ | $0.8922 \pm 0.0087$ | $0.3287 \pm 0.013$ | $0.0342 \pm 0.0066$ |
| | | DT | $0.7938 \pm 0.0574$ | $0.7956 \pm 0.0163$ | $0.191 \pm 0.0321$ | $0.0342 \pm 0.0066$ |
| | | XGB | $0.7847 \pm 0.0338$ | $0.8842 \pm 0.0081$ | $0.3092 \pm 0.0118$ | $0.0342 \pm 0.0066$ |
| | Adult Income | LR | $0.2288 \pm 0.0289$ | $0.8325 \pm 0.0044$ | $0.1903 \pm 0.0143$ | $0.0623 \pm 0.0084$ |
| | | DT | $0.4163 \pm 0.0473$ | $0.7588 \pm 0.0068$ | $0.1185 \pm 0.018$ | $0.0623 \pm 0.0084$ |
| | | XGB | $0.2736 \pm 0.0786$ | $0.8187 \pm 0.007$ | $0.143 \pm 0.0301$ | $0.0623 \pm 0.0084$ |
| | German Credit | LR | $0.9593 \pm 0.1347$ | $0.705 \pm 0.0302$ | $0.107 \pm 0.06$ | $0.0454 \pm 0.0322$ |
| | | DT | $0.9508 \pm 0.1584$ | $0.613 \pm 0.0294$ | $0.0543 \pm 0.0356$ | $0.0454 \pm 0.0322$ |
| | | XGB | $0.9028 \pm 0.2125$ | $0.672 \pm 0.0318$ | $0.0879 \pm 0.0589$ | $0.0454 \pm 0.0322$ |
| 1.0 | Law School | LR | $1.0 \pm 0.0$ | $0.9004 \pm 0.0002$ | $0.0005 \pm 0.0006$ | $0.001 \pm 0.001$ |
| | | DT | $0.9743 \pm 0.1191$ | $0.7268 \pm 0.0917$ | $0.0112 \pm 0.0197$ | $0.001 \pm 0.001$ |
| | | XGB | $0.974 \pm 0.1044$ | $0.8075 \pm 0.1055$ | $0.0105 \pm 0.0177$ | $0.001 \pm 0.001$ |
| | Diabetes | LR | $0.7826 \pm 0.1003$ | $0.6176 \pm 0.0075$ | $0.0714 \pm 0.0323$ | $0.0032 \pm 0.0022$ |
| | | DT | $0.979 \pm 0.0341$ | $0.5263 \pm 0.0071$ | $0.0076 \pm 0.0051$ | $0.0032 \pm 0.0022$ |
| | | XGB | $0.8584 \pm 0.0499$ | $0.5904 \pm 0.0121$ | $0.0481 \pm 0.0249$ | $0.0032 \pm 0.0022$ |
| | COMPAS | LR | $0.8046 \pm 0.0274$ | $0.8907 \pm 0.0077$ | $0.3229 \pm 0.0137$ | $0.0294 \pm 0.007$ |
| | | DT | $0.867 \pm 0.1809$ | $0.7369 \pm 0.0751$ | $0.1364 \pm 0.0681$ | $0.0294 \pm 0.007$ |
| | | XGB | $0.7476 \pm 0.0509$ | $0.8652 \pm 0.0306$ | $0.2796 \pm 0.0526$ | $0.0294 \pm 0.007$ |
| | Adult Income | LR | $1.0 \pm 0.0$ | $0.7607 \pm 0.0001$ | $0.0 \pm 0.0$ | $0.0046 \pm 0.0016$ |
| | | DT | $0.8519 \pm 0.1714$ | $0.6537 \pm 0.0195$ | $0.0013 \pm 0.0009$ | $0.0046 \pm 0.0016$ |
| | | XGB | $0.6111 \pm 0.167$ | $0.7467 \pm 0.0185$ | $0.0018 \pm 0.0014$ | $0.0046 \pm 0.0016$ |
| | German Credit | LR | $0.969 \pm 0.0821$ | $0.705 \pm 0.013$ | $0.0411 \pm 0.0585$ | $0.0385 \pm 0.0281$ |
| | | DT | $0.9004 \pm 0.0703$ | $0.617 \pm 0.0479$ | $0.0315 \pm 0.0402$ | $0.0385 \pm 0.0281$ |
| | | XGB | $0.9514 \pm 0.0972$ | $0.638 \pm 0.0429$ | $0.0435 \pm 0.0566$ | $0.0385 \pm 0.0281$ |
| 2.0 | Law School | LR | $1.0 \pm 0.0$ | $0.9004 \pm 0.0002$ | $0.0006 \pm 0.0005$ | $0.0012 \pm 0.0011$ |
| | | DT | $0.9792 \pm 0.0431$ | $0.7613 \pm 0.1223$ | $0.0012 \pm 0.0014$ | $0.0012 \pm 0.0011$ |
| | | XGB | $0.9651 \pm 0.0844$ | $0.814 \pm 0.1163$ | $0.0043 \pm 0.0071$ | $0.0012 \pm 0.0011$ |
| | Diabetes | LR | $1.0 \pm 0.0$ | $0.601 \pm 0.0$ | $0.0 \pm 0.0$ | $0.0013 \pm 0.0011$ |
| | | DT | $0.9924 \pm 0.0309$ | $0.5161 \pm 0.011$ | $0.0006 \pm 0.0005$ | $0.0013 \pm 0.0011$ |
| | | XGB | $0.9212 \pm 0.0581$ | $0.5748 \pm 0.0057$ | $0.0036 \pm 0.0028$ | $0.0013 \pm 0.0011$ |
| | COMPAS | LR | $0.8047 \pm 0.0275$ | $0.892 \pm 0.0094$ | $0.323 \pm 0.0116$ | $0.0296 \pm 0.006$ |
| | | DT | $0.9212 \pm 0.1419$ | $0.6431 \pm 0.0979$ | $0.0738 \pm 0.0634$ | $0.0296 \pm 0.006$ |
| | | XGB | $0.7556 \pm 0.0383$ | $0.8724 \pm 0.0184$ | $0.2899 \pm 0.0268$ | $0.0296 \pm 0.006$ |

Table 5: ($\mathcal{B}$ - Fair Predictions: regression) Similar table as Table 4 is provided for the regression task on the Communities and Crimes Dataset.

| $\eta$ | Model | KS | MSE | $Cov^2(r(X), \hat{Y})$ | $Cov^2(r(X), S)$ |
|---|---|---|---|---|---|
| | LR | $0.8223 \pm 0.0349$ | $0.0341 \pm 0.0046$ | $0.658 \pm 0.2337$ | $0.3462 \pm 0.1179$ |
| 0.0 | DT | $0.6653 \pm 0.0695$ | $0.0401 \pm 0.0068$ | $0.4062 \pm 0.0985$ | $0.3462 \pm 0.1179$ |
| | XGB | $0.7568 \pm 0.0259$ | $0.0254 \pm 0.003$ | $0.4376 \pm 0.1382$ | $0.3462 \pm 0.1179$ |
| | LR | $0.7047 \pm 0.0594$ | $0.0278 \pm 0.0032$ | $0.2645 \pm 0.0711$ | $0.2342 \pm 0.1327$ |
| 1.0 | DT | $0.5395 \pm 0.0549$ | $0.0509 \pm 0.0062$ | $0.2534 \pm 0.0624$ | $0.2342 \pm 0.1327$ |
| | XGB | $0.6668 \pm 0.0538$ | $0.032 \pm 0.0037$ | $0.2337 \pm 0.0571$ | $0.2342 \pm 0.1327$ |
| | LR | $0.7059 \pm 0.0595$ | $0.0277 \pm 0.0032$ | $0.2652 \pm 0.0706$ | $0.2345 \pm 0.1267$ |
| 2.0 | DT | $0.5934 \pm 0.0813$ | $0.057 \pm 0.0133$ | $0.2442 \pm 0.0973$ | $0.2345 \pm 0.1267$ |
| | XGB | $0.6666 \pm 0.0348$ | $0.0325 \pm 0.0025$ | $0.2435 \pm 0.0558$ | $0.2345 \pm 0.1267$ |

Figure 4: ($\mathcal{B}$ - Fair Predictions) Similar Figure as 1 for Law School and COMPAS datasets. In this case, we measured the fairness of the predictions made with the new representation in terms of the Equality of Opportunity (EOpp), which is represented versus the Accuracy. EOpp is estimated as the ratio $\hat{P}(c(X) = 1|S = 0, Y = 1)/\hat{P}(c(X) = 1|S = 1, Y = 1)$

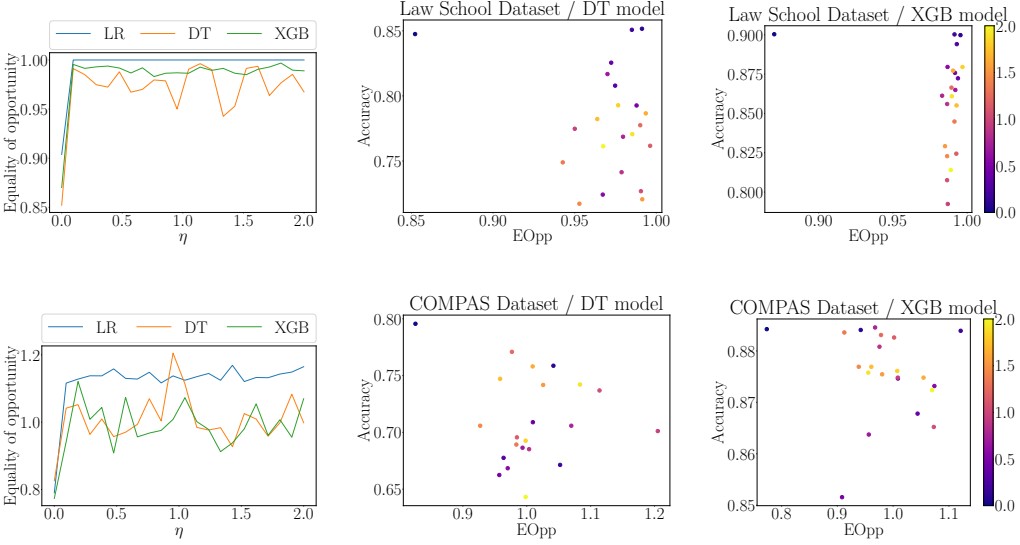

## B.4 COMPARISON WITH VANILLA FAIR PLS

We compared our proposed algorithm to the state-of-the-art method Vanilla Fair PLS Champion et al. (2023). The Vanilla PLS method consists in selecting the features that are related to the target (PLS) such that the correlation with the sensitive ones is below a predefined threshold $\tau$. In other words, it is a naive strategy that directly make use of the latent variables $(\mathbf{t}_1, ..., \mathbf{t}_k)$ generated with the standard PLS technique which are highly correlated with the outcome $\mathbf{Y}$ to impose fairness. This methodology is based on the conditional marginal distribution of those components to the sensitive variable $S$. In contrast, our formulation aims to obtain components that seek a balance between being target-related ($\eta$ small) and being independent with respect to the sensitive attribute ($\eta$ large enough). The left column of Figure 5 shows the behaviour of $\mathbf{C}^2_{r(\mathbf{X}),\mathbf{Y}}$ for new representations $r(\mathbf{X})$ obtained with the Vanilla PLS procedures. It is clear that, on the one hand, for $\tau$ close to 0 the representation has no features ($k = 0$). Moreover, as $\tau$ increases, the number of features in the representation also increases, while there is no trade-off between covariance with respect to the target $Y$, nor with respect to the sensitive feature. This is, the $\mathbf{C}^2_{r(\mathbf{X}),\mathbf{Y}}$ increases but the $\mathbf{C}^2_{r(\mathbf{X}),S}$

also increases, therefore fairness goal is not achieved. In contrast, this is not the case with Fair PLS as shown in the right column of Figure 5

Figure 5: Comparison of the covariance with respect to the target $Y$ and the sensitive attribute $S$ between the new representation $r(\mathbf{X})$ obtained via the Vanilla Fair PLS and our proposed formulation. The plots display the mean and standard deviation resulting from a 5-fold cross-validation procedure. For $\tau < 0.2$, $r(\mathbf{X}) \in \mathbb{R}^{1000 \times 0}$; for $0.2 \leq \tau < 0.6$, $r(\mathbf{X}) \in \mathbb{R}^{1000 \times 6}$ and for $0.6 \leq \tau < 1.0$, $r(\mathbf{X}) \in \mathbb{R}^{1000 \times 7}$. The data used for this analysis is described in Appendix B. 1. - Synthetic dataset.

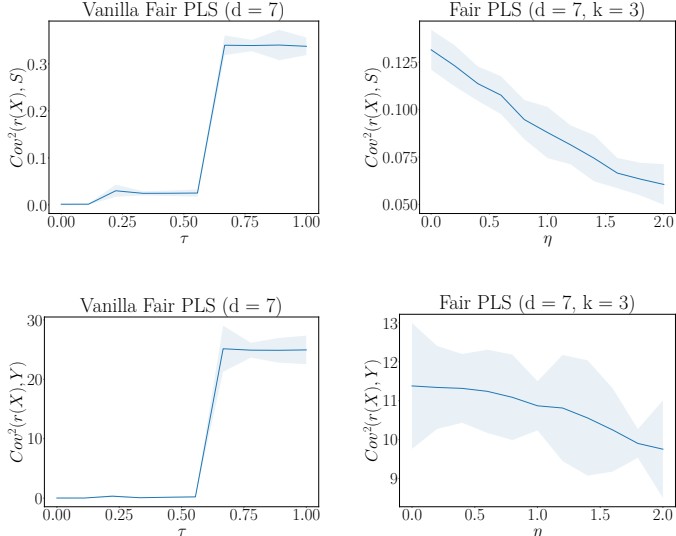

## B.5 COMPARISON WITH EXISTING METHODS FOR FAIR PCA

We compared our proposed algorithm by means of bias mitigation to the state-of-the-art Fair PCA method introduced by Kleindessner et al. (2023). First, we demonstrated that our method, like the aforementioned, manage to equalise the conditional means of the groups (see Figure 6). This is because the condition on the weights that states $Cov(\mathbf{w}^\mathsf{T}\mathbf{x}_i, s_i) = 0$, $\forall i \in [n]$, is equivalent to finding the optimal projection such that the group-conditional means of the projected data align.

Figure 6: Results of applying the Fair PLS method to the synthetic dataset from Kleindessner et al. (2023). Points in red color red are from group $S = 1$ and in blue color from group $S = 0$.

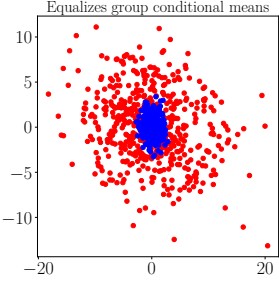

Secondly, we compared the performance of the two dimensional reduction techniques for Fair Representation Learning, Fair PCA (Kleindessner et al., 2023) and Fair PLS (ours), using the Adult Income dataset. This comparison consists on learning a fair classifier (Logistic Regression or Decision Tree Classifier) on top of the representation and measuring the classifier's predictive performance. While both methods achieve full fairness in terms of Demographic Parity ($DI = 1$) when being

applied as preprocessing methods, the predictive performance of Fair PCA is much lower than with Fair PLS (see Figure 7). Precisely, the mean accuracy for 7-fold CV is $0.7649$ for the Logistic Regression and $0.7480$ for the Decision Tree Classifier, while for Fair-PLS is higher than $0.8300$ and $0.7800$, respectively.

Figure 7: Comparison between the performance of Fair PCA and Fair PLS using Adult Income and Diabetes datasets. The points are test values of a 7-fold CV procedure. We have fixed $k = 2$ for both methodologies.

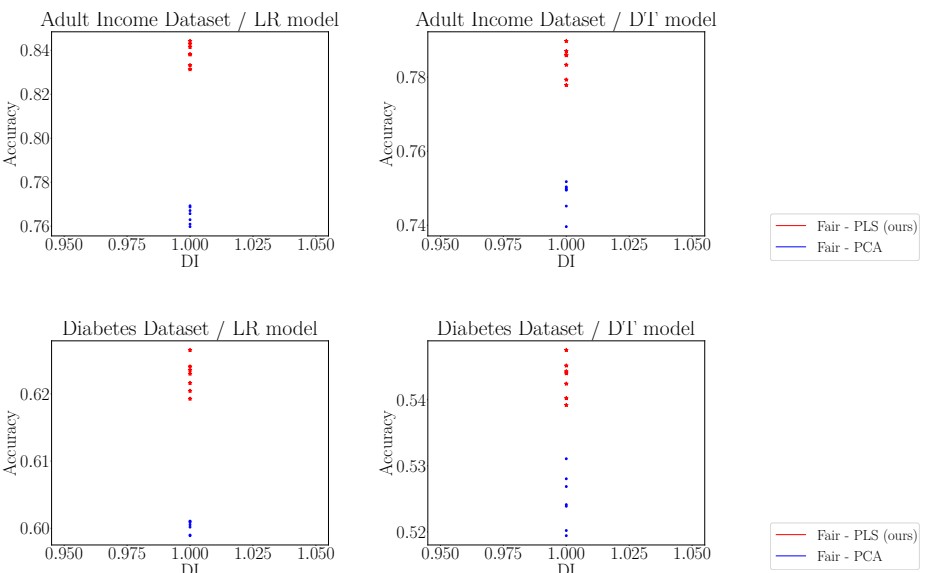

Notice that PCA works well because the orthogonality of the singular vectors eliminates the multicolinearity problem. But the optimum subset of components were originally chosen to explain $\mathbf{X}$ rather than $Y$, and so, nothing guarantees that the principal components, which 'explain' $\mathbf{X}$ optimally, will be relevant for the prediction of $Y$. The PCA unsupervised dimensionality reduction technique is based on the covariance matrix $\mathbf{X}^{\top}\mathbf{X}$. Nevertheless, in many applications it is important to weight the covariance matrix, this is, to replace $\mathbf{X}^{\top}\mathbf{X}$ with $\mathbf{X}^{\top}\mathbf{V}\mathbf{X}$, being $\mathbf{V}$ a positive definite matrix. PLS algorithm choose as $\mathbf{V}$ the representative matrix for the "size" of the data in the $\mathbf{Y}$ matrix, which is $\mathbf{V} = \mathbf{Y}\mathbf{Y}^{\top}$.

## B.6 RUNTIME COMPARISON

We tested our method in terms of running time of training with different data dimension and compared it with the standard PLS implementation of Scikit-learn. We used the data for this study provided by Lee et al. (2022b) as Synthetic data #2 In detail, the dataset is composed of two groups, each of half of the size $n$ and sampled from two different 3-variate normal distributions.

Figure 8: The runtime of the standard method, as implemented in (Pedregosa et al., 2011), is shown as a function of the data dimension in blue, while the corresponding runtime of the Fair PLS method is depicted in orange. The target dimension is fixed to 1.

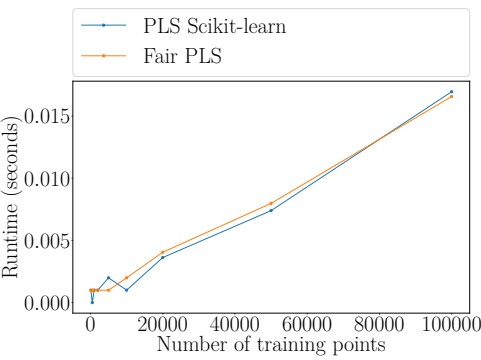

