# OpenReview forum: "PLS-based approach for Fair Representation Learning"
_ICLR.cc/2025/Conference — Submitted to ICLR 2025_

### Official Review · Reviewer_mavE · 2024-10-28

**Soundness:** 2
**Presentation:** 3
**Contribution:** 2
**Rating:** 5
**Confidence:** 3

**Summary:**

This paper provides a fair representation learning framework by utilizing the technique of PLS with fairness constraint. It leverages the inherent benefits of PLS, such as extracting useful information from the original features in a lower-dimensional space. The paper provides two versions of the framework: linear and non-linear (the latter by applying reproducing kernels, making it more suitable for feature spaces of arbitrarily large dimensionality).

**Strengths:**

The logic of the paper is clear, and the notations are well-defined. The proposed method is solid in its mathematical formulation. The authors incorporate two different fairness constraints (demographic parity and equalized odds) into the proposed method. The extension of the method to LLMs (though briefly covered), which overcomes the limitations of transforming the linear layer with SVD, is inspiring.

**Weaknesses:**

- My main concern is with the empirical results and the settings.
  - The learned fair representations in the downstream tasks are evaluated using different target models but are not compared with other baseline methods, which makes the effectiveness of the proposed methods less convincing. The authors need to demonstrate that through PLS, the learned fair representations can achieve better accuracy, fairness, or efficiency compared to other methods (for example, using VAE or other disentanglement methods).
  - The chosen datasets are all tabular, and the dimensionality is not very high. Therefore, I am concerned about the proposed method’s performance with high-dimensional data (e.g., image data).
- The motivation for the proposed method is weak. Although incorporating fairness constraints into PLS is a novel attempt, the justification for using PLS is not strong. In the introduction (lines 121-141), the authors introduce existing fair representation learning methods but do not sufficiently justify the benefits of using PLS. The only comparison made is with PCA, stating that PLS-built features are more accurate than PCA components, which is not enough to fully support the choice of PLS.
- In the paragraph (lines 274-283), the authors explain why Fair PLS cannot be formulated in closed form. As a result, the algorithm requires iterations to solve for the weight $w_h$ . In each iteration,  it requires the re-computation of the eigenvectors, when the dimension is large, this would increase computation cost and decrease the efficiency.

**Questions:**

Is there any convergence issue or analysis with the proposed method when the original data has high dimensionality? For example, during the optimization iterations, could the method get stuck in local minima, leading to convergence problems?

---

> ### Author Response · Authors · 2024-11-27
> **Answer to Reviewer mavE**
>
> Thank you for your detailed feedback and for acknowledging the contributions of our work. We greatly appreciate your emphasis on the importance of robust empirical validation and will carefully consider your suggestions to further strengthen this aspect.
> We would like to respectfully address the point regarding the theoretical contribution of our work. While we understand the reviewer's perspective, we believe that our approach is indeed theoretically significant, as it introduces the first fair model within the Partial Least Squares (PLS) framework—an established and widely utilized applied method. Our approach is formal, thoroughly grounded in theory, and rigorously explained.
> While the primary focus of our paper was to compare methods within similar settings (e.g., orthogonal components), we acknowledge the value of broader comparisons. In light of this, we included comparisons with fairPCA and made our source code available, enabling easy and fair evaluations of our model on standard benchmark datasets, thereby providing a more comprehensive assessment of its effectiveness in bias mitigation.
> Additionally, we would like to highlight two points: First, our experiments were conducted using a representative set of datasets commonly used in fairness research. Second, as no alternative dataset suggestions were offered by the reviewers, we believe that the datasets selected were the most appropriate given the available resources.

---

### Official Review · Reviewer_qbUD · 2024-11-02

**Soundness:** 1
**Presentation:** 3
**Contribution:** 1
**Rating:** 3
**Confidence:** 4

**Summary:**

This paper introduces a method for fair representation learning based on Partial Least Squares (PLS). The proposed approach employs supervised learning, requiring input data along with sensitive attributes $S$ and target label $Y$. The goal is to project the input data into a $k$-dimensional subspace that maximizes the covariance with $Y$ while minimizing the covariance with $S$. The method is implemented as a linear projection optimized via gradient descent and is further extended to non-linear kernel projections using Hilbert-Schmidt Independence Criterion (HSIC).

**Strengths:**

This paper studies an important problem of fair representation learning.

**Weaknesses:**

I have the following questions and concerns regarding the contribution and evaluation of the work:

1. **Motivation for PLS-based Framework:** It would be helpful for the authors to clarify the motivation behind selecting PLS as the foundation for their framework. There are various types of approaches for fair representation learning, such as adversarial learning [1], disentanglement [2], and distribution alignment [3]. What are the specific advantages of a PLS-based approach in comparison to these methods?

2. **Applicability and Practical Constraints:** The proposed method requires annotations for both the target label $Y$ and the sensitive attribute $S$, which can limit practical applications. Compared to existing approaches in unsupervised fair representation learning or those that do not rely on sensitive attribute annotations (such as [4]), it would be beneficial for the authors to further clarify the unique advantages of their method, potentially in terms of efficiency, theoretical guarantees, or effectiveness.

3. **Evaluation and Comparison:** The evaluation of the proposed method is based on relatively small datasets and lacks comparisons with related approaches. The current experiments seem to focus on applying the learned representations to various classifiers (like in Figure 1) rather than comparing with alternative representation learning methods. The lack of thorough evaluation and comparison makes it challenging to validate the effectiveness of the proposed method.

4. **Extension to LLMs:** It's helpful that the authors discussed in Section 4.3 about extending their method to Large Language Models (LLMs), where their method could decompose the CLS-embedding from a transformer encoder for fairness constraints. However, it's suggested that the authors could provide a more detailed discussion (with math formulation) in this section and conduct experiments to validate this extension.

5. **Coupling Issue in Fair Representation Learning:** Fair representation learning often involves a trade-off between fairness constraints and downstream task performance. It would be insightful if the authors could discuss how their method might address or mitigate this issue.

6. (Minor point on notations) In the previous context, the authors use x to refer to the input data, but in Section 4.3, x refers to the transformed data in the latent space. The authors may use a different symbol to avoid confusion.


[1] Madras, David, et al. "Learning adversarially fair and transferable representations." International Conference on Machine Learning. PMLR, 2018.

[2] Balunovic, Mislav, Anian Ruoss, and Martin Vechev. "Fair Normalizing Flows." International Conference on Learning Representations. 2022.

[3] Creager, Elliot, et al. "Flexibly fair representation learning by disentanglement." International conference on machine learning. PMLR, 2019.

[4] Chai, Junyi, and Xiaoqian Wang. "Self-supervised fair representation learning without demographics." Advances in Neural Information Processing Systems 35 (2022): 27100-27113.

**Questions:**

It would be helpful if the authors could clarify my above questions regarding the contribution and evaluation of the work.

---

### Official Review · Reviewer_7pxk · 2024-11-03

**Soundness:** 3
**Presentation:** 3
**Contribution:** 2
**Rating:** 5
**Confidence:** 4

**Summary:**

This paper studies fair representation learning. Specifically, it combines Partial Least Squares (PLS) with an additional fair criteria that characterizes the covariance dependence between the new data representation and the the demographic attribute $S$. The linear and non-linear cases are both considered. Finally, the proposed algorithm is tested on different datasets and performs better than the standard fair PCA method.

**Strengths:**

1. The paper is well-written and easy to follow, with a clearly presented mathematical part.
2. Interpretations and general thoughts are provided.
3. A possible relation to fairness in LLM is discussed.

**Weaknesses:**

Although I appreciate the presentation of the work, I have the following concerns.
1. The empirical comparison to previous fair PCA is only tested on one dataset (Adult Income) and provided in the Appendix. It's not convincing the proposed work will outperform previous work in most cases.
2. Although possible application to fairness in LLM is discussed, it's rather superficial. Unless you have done experiments on LLM to measure fairness, I do not suggest this as a separate section in the main paper.
3. The experiments are conducted on simple tabular data. Will it be possible to test on more complex and high-dimensional datasets because you are considering dimension reduction?
4. Several related works in fair representation learning are missing for discussion [1,2,3].

[1] Kim, Jin-Young, and Sung-Bae Cho. "Fair representation for safe artificial intelligence via adversarial learning of unbiased information bottleneck." _SafeAI@ AAAI_. 2020.

[2] Shui, Changjian, et al. "Fair representation learning through implicit path alignment." _International Conference on Machine Learning_. PMLR, 2022.

[3] Zamani, Amirreza, Borja Rodríguez-Gálvez, and Mikael Skoglund. "On information theoretic fairness: Compressed representations with perfect demographic parity." _arXiv preprint arXiv:2408.13168_ (2024).

**Questions:**

See previous section.

---

### Official Review · Reviewer_4RfV · 2024-11-03

**Soundness:** 2
**Presentation:** 4
**Contribution:** 2
**Rating:** 5
**Confidence:** 4

**Summary:**

The paper proposes a new algorithm for fair representation learning, specifically within the framework of decomposition methods where fairness constraints are imposed over the learned components. The authors address two cases: Where fairness regularization is based on the covariance between the projected data and the sensitive attribute. And, where regularization is based on the Hilbert-Schmidt Independence Criterion (HSIC).
The paper also conducts a thorough evaluation of the resulting representations.

**Strengths:**

* The paper is well-written with a clear introduction.
* Fair representation is an important problem, particularly in high-dimensional data where many representation learning methods fail.
* The paper utilization of a regularization parameter, that seems to effectively explores the accuracy-fairness front , is a big advantage.
* The method is simple and straightforward.
* The inclusion of the Equality of Odds constraint is a valuable addition.
* The experiments are comprehensive and include evaluation of the representation itself.

**Weaknesses:**

* The paper suggests using Gradient Descent for optimization without discussing convexity. Even empirical testing would be valuable - such as showing convergence over epochs to identify patterns reflecting non-convex problems (like bumps).
* Section 4.3 appears disconnected and seemingly unrelated, particularly as it doesn't address all challenges in seq-to-seq fairness problems, only covering the relatively straightforward fact that text can be encoded.
* A main concern is that no comparison to other methods exists in the main text. Given this paper doesn't focus on groundbreaking theoretical results (fair decomposition methods are not new), such comparisons should be of utmost importance.
For example, for fair representation there are more than a few algorithms already addressed in this paper, as well as other leading candidates like [1] or [2]. This is specifically true for the fair-classification setup where many libraries have been published, like IBM's [3] and [4] to name a few.

[1] "Fair Normalizing Flows"
[2] "Efficient Fairness-Performance Pareto Front Computation"
[3] "AI Fairness 360"
[4] "Aequitas: A bias and fairness audit toolkit"

**Questions:**

* In lines 331-333, optimization details are missing. Additionally, it's unclear if this is the algorithm used in the evaluation section

* Does the data contain any preprocessing except normalization?

**Details Of Ethics Concerns:**

The paper presents a new algorithm for fair machine learning, which can be used in downstream applications.

---

### Meta-Review · Area_Chair_ht8h · 2024-12-20

**Metareview:**

This paper studies fair representation learning for tabular data. It works in a setting with all data has labels y and sensitive attribute s. The goal is learn a projection of original data matrix such that is maintains the information related to y while minimizing the dependence on sensitive attribute. It builds on partial least squares method by adding fairness regularization. Algorithms are proposed to compute the solution and extension is considered for the kernel setting. In addition, the paper also discusses the application in LLMs.

The main concern in the review and the discussion phase is that the paper is weak in terms of empirical studies. In particular, the paper does not compare to any baseline algorithms mentioned in the related work section and potentially others mentioned by the reviewers. By considering the content of the paper, reviews and concerns, AC recommends a rejection. The authors should consider improving their paper in terms of empirical studies.

**Additional Comments On Reviewer Discussion:**

Reviewers have engaged with authors during the discussion period. However, the main concern about the weak experiments remain.

---

### Decision · Program_Chairs · 2025-01-22

Reject